# Unlocking the Therapeutic Potential of Ellagitannins: A Comprehensive Review of Key Representatives

**DOI:** 10.3390/molecules30224328

**Published:** 2025-11-07

**Authors:** Rositsa Mihaylova, Viktoria Elincheva, Georgi Momekov, Rumyana Simeonova

**Affiliations:** Department of Pharmacology, Pharmacotherapy and Toxicology, Faculty of Pharmacy, Medical University of Sofia, 1000 Sofia, Bulgaria; v.lyubomirova@pharmfac.mu-sofia.bg (V.E.); gmomekov@pharmfac.mu-sofia.bg (G.M.)

**Keywords:** ellagitannins, urolithins, punicalagin, sanguiin H-6, geraniin, corilagin, oenothein B, chebulagic acid, chebulinic acid, anti-inflammation

## Abstract

The present review offers a comprehensive synthesis of the structural diversity, natural occurrence, and therapeutic promise of key ellagitannins (punicalagin, sanguiin H-6, corilagin, geraniin, oenothein B, chebulagic, and chebulinic acids) within the hydrolyzable ellagitannin pool. Distributed in medicinal and dietary plants long used in traditional medicine, ellagitannin-rich species serve as sources of both complex polyphenolic scaffolds and their bioactive metabolites, urolithins, which mediate many of their health-promoting effects. Special emphasis is placed on the multifaceted mechanisms that contribute to their potent antioxidant, anti-inflammatory, antimicrobial, and anticancer effects, extending to both non-communicable and communicable diseases. Despite their broad therapeutic spectrum, clinical translation is limited by challenges such as poor bioavailability, host-gut microbiota variability, and a lack of robust in vivo evidence. The review highlights future directions aimed at unlocking ellagitannins’ potential, including microbiota-targeted strategies for urolithin production, the design of stable prodrugs and analogs, and innovative delivery platforms. By integrating phytochemical, mechanistic and translational insights, this article positions ellagitannins as promising candidates for the development of novel polyphenol-based interventions.

## 1. Introduction

Polyphenols constitute one of the most abundant and chemically versatile classes of plant secondary metabolites, encompassing flavonoids, stilbenes, phenolic acids, and tannins [1,2]. Flavonoid polyphenols, including quercetin, catechins, and anthocyanins, are ubiquitous in fruits, vegetables, tea, and wine, and have been the subject of extensive scientific investigation for their roles in oxidative stress modulation, inflammation, and metabolic regulation through established cellular mechanisms [3]. As a result, these constituents dominate the phytochemical literature, with extensive experimental and clinical research on their well-documented health-promoting effects.

In contrast, the class of polyphenols belonging to the hydrolysable tannins class has been far less investigated, despite emerging evidence of their distinctive chemical features and bioactivities. As representative plant metabolites with abundant hydroxyl groups, particularly tannins of the hydrolysable type, have recently attracted growing attention for their emerging significance in human health and diseases [4].

Hydrolysable tannins are a diverse group of plant polyphenols, encompassing ellagitannins, gallotannins, and various types of complex tannins, distinguished by their structural complexity, considerable molecular weight, and multifaceted bioactivities. They belong to the broader class of tannins, which, as major members of the secondary metabolite repertoire, hold significant ecological, nutritional, and pharmacological relevance [1]. Unlike condensed tannins, hydrolysable tannins readily undergo hydrolysis under acidic or enzymatic conditions, yielding sugars (typically glucose) and phenolic acids such as gallic or ellagic acid and their derivatives [2,5].

Within the hydrolyzable tannin pool, a subset of plant-specific ellagitannin derivatives—punicalagin, sanguiin H-6, corilagin, geraniin, oenothein B, chebulagic and chebulinic acids, is increasingly recognized as a promising area of research, yet still receives comparatively limited mechanistic and translational investigation, despite accumulating evidence of biological relevance [4]. These compounds exhibit considerable structural complexity, encompassing composite structures with distinctive monomeric, macrocyclic, or dimeric architectures. Their occurrence in plants long valued in traditional medicine, including *Terminalia* species in Ayurveda, *Epilobium* in European and Asian phytotherapy, as well as dietary sources such as pomegranate and raspberries, asserts their ethnopharmacological importance [4,6].

The present review consolidates recent findings relating to the molecular diversity, natural distribution, and multifaceted bioactivities of key derivatives within the ellagitannin family, selected based on their structural diversity, prevalence in ethnomedicinal and dietary sources, and their established or emerging significance in pharmacological and mechanistic studies. Special emphasis is placed on elucidating the cellular and molecular mechanisms that mediate their potent antioxidant, anti-inflammatory, antimicrobial (including antibacterial, antifungal, antiviral), anticancer, cardiovascular, metabolic, and neuroprotective properties. The review also addresses critical barriers to the clinical translation of ellagitannins, including poor bioavailability, the scarcity of in vivo and clinical studies, and the underappreciated role of gut microbiota in the metabolism and bioactivation of these compounds. We further outline future research directions, focused on facilitating bioactivation through urolithin-producing probiotics (e.g., *Gordonibacter urolithinfaciens*), developing more stable and permeable prodrugs or molecular conjugates, engineering semi-synthetic analogs with enhanced lipophilicity and metabolic resilience, and employing technological delivery approaches.

## 2. Natural Sources and Ethnopharmacological Background

Ellagitannins represent an expansive and chemically diverse group of high-molecular-weight polyphenols, widely distributed throughout the plant kingdom in nearly all tissues of plants, including roots, leaves, bark, wood, galls, fruits, and seeds [1,4,7]. In their ecological context, ellagitannins act as key defensive metabolites, deterring herbivores and conferring resistance to microbial pathogens, while their ethnopharmacological relevance is substantiated by their deep integration into diverse traditional medicine systems. More recently, a growing body of research has pointed to their promising therapeutic potential, encouraging further investigation into their natural sources and diverse biological activities [8]. Nevertheless, the body of evidence supporting their biological and pharmacological properties is unevenly distributed among individual compounds. The most comprehensive data concern punicalagin and the gut microbiota-derived metabolites urolithins, which have been extensively studied in the contexts of food science, pharmacology, and human health. By contrast, other ellagitannins remain comparatively underexplored, with limited insight into their bioavailability, metabolism, and mechanistic effects. Understanding this gap is key to interpreting current evidence and directing future research toward the lesser-studied members of this class.

Punicalagin is among the most extensively characterized ellagitannins, found in high abundance in the pericarp, rind, bark and heartwood of *Punica granatum* (*pomegranate*), but also present in members of the Myrtales order, including the Terminalia (*T. myriocarpa*, *T. catappa*, *T. chebula*, *T. arjuna*) and Combretum genera (*Combretum molle*), distributed in Africa, Asia, Australia and the Americas [9,10,11]. Notably, in *T. chebula*, punicalagin co-occurs with the hallmark ellagitannins of the species, namely chebulagic and chebulinic acids, as well as corilagin (a gallotannin-ellagitannin hybrid), constituting the principal hydrolyzable tannin pool of this species [6,12,13]. Accordingly, *Terminalia chebula* (“haritaki”) is highly esteemed in Ayurvedic practice for its central role in the triphala formulation, credited with supporting digestive function, facilitating tissue regeneration, and aiding in the removal of toxins, effects largely attributed to its abundant tannin constituents [14]. Corilagin is also a common constituent of *Punica granatum* and pomegranate extracts, and of other medicinal species across the Euphorbiaceae (e.g., *Phyllanthus emblica*, *P. urinaria*, *P. tenellus*, *P. niruri*, *Acalypha australis*), Geraniaceae (*Geranium sibiricum*), Polygonaceae (*Polygonum chinense*), and Saururaceae (*Saururus chinensis*) families [15].

Geraniin, a dehydroellagitannin, is exceptionally abundant in *Phyllanthus* species, particularly *P. amarus*, *P. urinaria*, and *P. muellerianus*, traditionally used to manage menstrual disorders, wound infections, fevers, pain, and inflammation [16,17,18], and is also present in significant amounts within certain Geranium taxa (such as *G. bellum*, *G. wilfordii*, and *G. carolinianum*). Moreover, geraniin constitutes a principal bioactive compound in the traditional Chinese medicine Herba Geranii et Erodii, which is valued for its efficacy in rheumatism, joint pain, infectious diseases, and dysentery. Notably, hydrolytic cleavage of geraniin yields the ellagitannin corilagin, further enhancing its therapeutic scope [19,20,21].

Sanguiin H-6, representative of the *Rubus* genus (raspberry, blackberry), and oenothein B, a dimeric ellagitannin occurring in *Epilobium* spp., further exemplify this group’s phytochemical diversity [4]. Importantly, ellagitannins significantly contribute to the polyphenolic profile of edible plants and foods, including strawberries, blackberries, cranberries, walnuts, and hazelnuts, broadening their dietary relevance beyond classical medicinal species [1,22]. In European phytotherapy, *Epilobium* teas (oenothein B-rich) are traditionally used for urinary and prostate disorders, while oak gall and pomegranate extracts are utilized as topical astringents for oral, gingival, and cutaneous inflammatory conditions [4,23,24].

## 3. Structural Diversity and Structure-Activity Relationships (SARs) of Representative Ellagitannins

Tannins are structurally diverse polyphenols, generally classified into hydrolyzable tannins (including ellagitannins and gallotannins), condensed tannins, complex tannins (fusing hydrolysable and condensed motifs), and the less prevalent phlorotannins found in marine algae [25,26,27,28]. Condensed tannins, also known as proanthocyanidins, well-studied for their potent antioxidant and anti-inflammatory properties, are polymers of flavan-3-ol units that are resistant to hydrolysis [28].

Hydrolyzable tannins, on the other hand, are esters of a central sugar core (most commonly D-glucose) with gallic acid (in the lesser common gallotannins) or with oxidatively coupled galloyl units that form the hexahydroxydiphenoyl (HHDP) group, present in the widely distributed ellagitannins [25,26]. (Figure 1). The HHDP moieties typically undergo spontaneous lactonization, yielding ellagic acid upon hydrolysis, which is the chemical hallmark of ellagitannins [4,29]. This chemical distinction gives rise to pronounced differences in the physicochemical properties, metabolic fate, and biological properties of both subclasses; gallotannins are typically less complex and readily hydrolyzed to gallic acid, whereas ellagitannins, featuring macrocyclic or polymeric HHDP units, offer extensive structural variability and broad-spectrum bioactivities [25,29].

Reflecting both subclasses, corilagin’s molecular structure features both a galloyl and an HHDP units esterified to the same sugar backbone [30,31]. Its unique structure accounts for its diverse extraction profiles, flexible pathways for enzymatic hydrolysis, and the ability to functionally mimic either gallotannin or ellagitannin characteristics according to context [27].

Complex tannins (hydrolysable flavono-ellagitannins) also exhibit bifunctional properties, containing both galloyl and HHDP esters on the glucose core alongside flavan-3-ol motifs, endowing them with features of both hydrolyzable and condensed tannins [27]. Unlike C-glycosidic tannins, they occur in a limited number of plant families, including Combretaceae, Myrtaceae, Melastomataceae, Fagaceae, and Theaceae [32].

The ellagitannin family spans a broad structural continuum, ranging from relatively simple monoesters such as corilagin, through highly substituted monomeric forms exemplified by geraniin, to complex oligomeric and macrocyclic architectures such as sanguiins and oenothein B (Figure 2). This molecular diversity underlies their wide-ranging biochemical mechanisms and physiological effects [4,33,34]. Within this review’s focus on representative ellagitannins (punicalagin, sanguiin H-6, corilagin, geraniin, oenothein B, chebulagic acid, and chebulinic acid), structure-activity relationship (SAR) analyses reveal that the number and distribution of galloyl, HHDP, and chebuloyl moieties modulate their capacities for metal chelation, radical scavenging, enzyme modulation, and molecular target specificity [35,36].

The high density of phenolic hydroxyl groups in the HHDP residue often increases in vitro antioxidant metrics of ellagitannins; however, it renders them too large and polar to cross the intestinal epithelium in their native form. Polymerization and macrocyclic topology (i.e., oenothein B, sanguiin-H6) are also decisive for protein interactions and metal chelation: multiple adjacent catechol- and pyrogallol-type hydroxyls bind metal ions (Fe, Cu) strongly and provide the chelation sites implicated in both antioxidant activity and, paradoxically, pro-oxidant Fenton chemistry in some settings. The spatial arrangement of these hydroxyls, whether on galloyl appendages, within HHDP rings, or distributed across oligomeric scaffolds, determines chelation geometry and redox potential. For instance, the HHDP unit tends to form a constrained biaryl diester that can stabilize radical intermediates and coordinate metals in a fashion not reproducible by isolated galloyl esters; this partially explains why HHDP-containing ellagitannins often outperform simple gallic acid in cell-free antioxidant assays and why HHDP stereochemistry affects ferroptosis-modulating activity observed for chebulagic/chebulinic acids [33,34,35,37,38,39,40].

On the other hand, the principal in vivo biological activities of dietary ellagitannins are mediated by their hydrolytic degradation products, namely ellagic acid and, foremostly, their final dibenzopyran-6-one metabolites, called urolithins, generated through sequential transformation by the gut microbiota. The microbial processing typically unfolds through a stepwise dehydroxylation, with ellagic acid first converted into intermediate products (i.e., urolithin D and C), and ultimately producing the final metabolites urolithin A and B. Urolithin A, in particular, is recognized as the principal circulating metabolite conferring the anti-inflammatory, antioxidant, and organoprotective effects characteristic of dietary ellagitannins in vivo [25,41,42]. Notably, interindividual variability in the abundance of urolithin-producing bacterial taxa (including *Gordonibacter* spp.) results in distinct metabotypes (A, B, and non-producers 0), with only subsets of the population efficiently converting precursors to bioactive urolithins [25,41,43].

Vice versa, ellagic acid and its gut-derived urolithin metabolites play a pivotal role in modulating the intestinal microbiota, reshaping both microbial diversity and function. These interactions foster an environment that promotes beneficial bacteria and suppresses potentially harmful species, contributing to strengthened intestinal barrier function, elevated antioxidant capacity and reduced inflammation. Recent animal studies and clinical interventions have demonstrated that ellagitannin supplementation leads to marked modulation of gut microbiota composition. For example, administration of ellagic acid at 0.3% of the daily diet in rodent models for 21 days resulted in increased jejunal villus height, elevated relative abundance of beneficial taxa such as *Bacteroidetes*, *Lactobacillus*, *Ruminococcaceae* and *Clostridium ramosum*, and reductions in potentially harmful groups including *Firmicutes*, *Streptococcus*, and *Rothia* [44]. Furthermore, a recent clinical study demonstrated that walnut supplementation caused notable shifts in gut microbiota composition, as revealed by 16S rRNA gene sequencing. Both beta diversity and Shannon’s alpha-diversity were significantly altered after walnut intake, independent of demographic factors and baseline urolithin metabotypes. Several bacterial genera showed differential enrichment pre- and post-supplementation; importantly, the abundance of *Gordonibacter*, a genus capable of producing urolithin A from ellagic acid, increased following walnut consumption, suggesting a link between dietary ellagitannin intake and enhancement of bacteria involved in urolithin production [45]. These microbial shifts strongly suggest that ellagitannin consumption can reprogram community composition toward a healthier profile, fostering a host environment characterized by increased production of protective short-chain fatty acids and metabolites, creating biochemical conditions that help counter the development and progression of cancer. For example, rodent studies administering ellagic acid or pomegranate extract (a rich source of ellagitannins) have shown decreased incidence and multiplicity of colon tumors, coinciding with increased abundance of urolithin-producing bacteria, lower inflammatory markers and improved gut integrity [25,46,47]. Other human intervention studies reveal similar trends: individuals with higher capacity for microbiome-mediated urolithin production exhibit greater systemic and colonic exposure to these metabolites and are hypothesized to derive enhanced chemopreventive benefits from ellagitannin-rich diets or supplements [47].

Punicalagin, a hallmark constituent of pomegranate, is among the highest molecular-weight monomeric ellagitannins identified in food, with a molecular weight of 1084.72 g/mol [48]. Importantly, it is among the few dietary ellagitannins that directly incorporate the actual dilactone ellagic acid moiety within their structure, rather than forming it upon hydrolysis. Modern structural revisions have reaffirmed its highly symmetrical and rigid macrocyclic dimeric nature, contributing to its known antioxidant, anti-inflammatory, anti-adipogenic, and anticancer activities. The SAR of punicalagin has been demonstrated to rely on the stereochemistry and spatial proximity of the HHDP and gallagyl groups, with maximal protein- and metal-binding afforded by the macrocyclic arrangements [11,49]. As with all high-molecular-weight ellagitannins, its solubility is low and membrane permeability is poor, accounting for rapid intestinal hydrolysis and decomposition to ellagic acid and, subsequently, urolithin [50].

Sanguiin H-6 is a dimeric casuarictin derivative consisting of repeating galloyl-glucose-HHDP units, and a major macrocyclic, polymeric ellagitannin in the Rosaceae and Rubus genera. The dimeric topology increases rigidity and yields high avidity for nucleic acids and nuclear enzymes (demonstrated as topoisomerase inhibition), potent in vitro cytotoxicity against transformed cell lines, and exceptional affinity for serum albumin and metal ions [51,52,53].

Corilagin, widely distributed in *Phyllanthus* and *Terminalia* species, is recognized for its hybrid structure exhibiting features of both gallotannins and ellagitannins: a gallic acid ester at one glucose hydroxyl, and a single HHDP at two others [31,54]. The SAR of corilagin is dominated by the capacity of both galloyl and HHDP groups to chelate metals, regulate enzyme activity (notably of pro-inflammatory COX and LOX isoforms), and scavenge ROS [55]. Despite its smaller molecular size, absorption studies show that corilagin itself is poorly transported across intestinal epithelia and displays enhanced hydrolytic breakdown compared to larger macrocyclic and polymeric ellagitannins [56]. Importantly, corilagin is also a typical hydrolysis product of larger ellagitannins (notably geraniin), which justifies its occurrence in decoctions and processed plant extracts.

Geraniin is paradigmatic of monomeric, highly substituted ellagitannins whose chemical lability determines both analytical behavior and biological fate: containing galloyl, hexahydroxydiphenoyl (HHDP) and dehydrohexahydroxydiphenoyl (DHHDP) groups, characteristic of ellagitannins, geraniin readily undergoes partial hydrolysis (e.g., on heating or in acidic media) to generate corilagin, brevifolin derivatives, gallic and ellagic acids—metabolites that have their own bioactivities and that complicate interpretation of crude-extract pharmacology. The hydrolysis is facilitated by the strategic positioning of the HHDP and DHHDP esters, both susceptible to enzymatic and chemical cleavage, particularly in the GI tract or during extended decoction. The hydrolytic susceptibility of geraniin also affects formulation choices and extraction yields and is a mechanistic basis for the frequent observation that decoctions or heated extracts display different biological profiles than cold extractions [57].

Oenothein B occupies a special place because it is a macrocyclic dimeric ellagitannin with a rigid, cyclic architecture that markedly alters its pharmacodynamics relative to linear monomers. Macrocyclization substantially increases molecular rigidity and creates a large, multivalent surface for interacting with proteins and receptors; accordingly, oenothein B exerts robust immunomodulatory effects (activation of phagocytes, modulation of cytokine release, and effects on innate lymphocytes), distinct enzyme-inhibitory profiles, and pronounced antioxidant capacity in vitro and in vivo. However, like other large ellagitannins, it is poorly permeable as an intact molecule and its in vivo actions likely reflect a combination of local (gastrointestinal) effects, modulation of microbiota, and generation of lower-molecular-weight metabolites [58].

Chebulagic and chebulinic acids, the most abundant hydrolyzable tannins in *Terminalia* fruits, exemplify how the stereochemistry of key moieties fundamentally shapes both chemical and biological profiles. Both compounds share a β-D-glucose backbone glycosylated with chebuloyl and galloyl esters, but structural differences arise from the number and arrangement of these groups, and especially from the absolute configuration and spatial orientation of the chiral axis in the HHDP motif [59]. Chebulagic acid incorporates a single galloyl, one chebuloyl, and one HHDP group, the latter two forming a macrocyclic (R)-configured ring, which imparts greater rigidity and spatial hindrance, translating to distinct enzyme affinities and kinetic properties. In contrast, chebulinic acid features three galloyl and one chebuloyl groups and adopts a more flexible “skew-boat” conformation, facilitating broader accessibility to active sites and more diverse binding interactions with biological targets. [37,59].

## 4. Biological Activities and Molecular Mechanisms of Ellagitannins

Despite the growing interest in the diverse class of ellagitannins, there is a pronounced disparity in the amount of experimental and clinical evidence available for different members of this group. The vast majority of mechanistic and translational studies, as well as published clinical trials, focus on punicalagin, ellagic acid, and urolithin derivatives found predominantly in pomegranate and widely investigated for their pharmacokinetics, bioactivities, and formulation strategies. In contrast, far fewer studies have examined the biological effects, biotransformation, or potential applications of other representative ellagitannins such as sanguiin H-6, corilagin, geraniin, oenothein B, chebulagic acid, and chebulinic acid (Table 1). Acknowledging these research gaps is essential for balanced interpretation and fostering future investigations that address the untapped potential of lesser-studied ellagitannin structures within this multifunctional compound class.

### 4.1. Antioxidant and Anti-Inflammatory Activities

Interception of oxidative stress and inflammatory signaling is fundamental to the health-protective effects of dietary polyphenols. Chronic oxidative damage, propelled by reactive oxygen species (ROS), underlies numerous human diseases, including cancer, cardiovascular, and inflammatory pathologies [60]. In vitro studies reveal that high-molecular-weight polyphenols, including ellagitannins, often display superior antioxidant potency compared to smaller phenolics and reference antioxidants such as ascorbic acid and α-tocopherol [60,61,62]. Their remarkable radical-scavenging ability enables disruption of free radical chain reactions, preventing lipid, protein, and DNA oxidative damage. Building on this, it is important to highlight that individual members of this group exhibit distinct yet complementary antioxidant and anti-inflammatory effects. Their complex polyphenolic structures confer the ability not only to neutralize free radicals directly but also to chelate transition metals and modulate redox-sensitive transcription factors. By attenuating oxidative stress at the molecular level, they further interrupt the activation of pro-inflammatory signaling cascades, including NF-κB and MAPK pathways, thereby reducing the production of cytokines, chemokines, and inflammatory enzymes. Furthermore, some ellagitannin derivatives can directly suppress key enzymes related to inflammation (e.g., COX and LOX), contributing to their anti-inflammatory profile. These multifaceted mechanisms underpin the therapeutic potential of ellagitannins in the prevention and management of inflammatory, cardiovascular, and neoplastic disorders [3,25,62].

A prime illustration of these bioactivities can be found in punicalagin, whose molecular characteristics and abundance in pomegranate make it a key subject in ellagitannin research. It demonstrates powerful antioxidant and anti-inflammatory activities, effectively neutralizing superoxide anions, hydroxyl radicals, and peroxynitrite while chelating transition metals and inhibiting lipid peroxidation, thereby protecting cellular membranes and nucleic acids. Numerous studies demonstrate its ability to upregulate endogenous antioxidant enzymes like superoxide dismutase (SOD) and catalase, thereby enhancing cellular defense mechanisms [50,63]. Punicalagin’s anti-inflammatory effect is underscored by inhibition of the NF-κB pathway, suppression of pro-inflammatory cytokines such as TNF-α, IL-6, IL-1β, and downregulation of COX-2 and iNOS—molecular actions that support its therapeutic potential in inflammatory diseases [11,49]. Importantly, it has been shown to inhibit vascular inflammation by reducing the expression of downstream pro-inflammatory cytokines and adhesion molecules (ICAM-1, VCAM-1), and blocking VEGF-induced endothelial signaling pathways [64].

Among sanguiins, sanguiin H-6 (SH6) is frequently identified as the principal compound responsible for antioxidant activity. Its antioxidant potential has been demonstrated in models of oxidative stress, such as lipopolysaccharide (LPS)-induced peroxynitrite (ONOO^−^) production and ischemia–reperfusion injury, where SH6 pre-treatment reduced ONOO^−^-mediated damage and improved kidney function [65]. Additionally, ellagitannins derived from Rubus berries, including dimeric forms such as SH6 and SH10, have been shown to act as effective radical scavengers in DPPH assays and inhibit lipid peroxidation in both bulk and emulsified methyl linoleate systems, as well as in human low-density lipoprotein in vitro [66]. SH6 significantly suppressed cytokine-induced neutrophil migration without exhibiting cytotoxicity toward neutrophils [67]. Furthermore, at a concentration of 2.5 μM, ellagitannin treatment completely inhibited both TNF-α- and IL-1β-induced IL-8 release and suppressed NF-κB transcriptional activity [68]. In another study, Sanguiin H-6 potently suppressed nitric oxide generation in a concentration-dependent manner and significantly downregulated the activity of inducible nitric oxide synthase (iNOS), preserving cellular viability. In addition, it effectively scavenged nitric oxide derived from sodium nitroprusside—a nitric oxide-releasing compound, demonstrating both inhibitory and direct neutralizing actions against NO-mediated cellular stress [69].

Corilagin, a prototypical gallotannin-ellagitannin hybrid, also possesses broad-spectrum antioxidant and anti-inflammatory activities. As demonstrated in both cellular and animal models, it acts as a powerful scavenger of hydroxyl and nitric oxide radicals, surpassing standard antioxidants in LDL and lipid protection assays, and effectively inhibits xanthine oxidase [31,70,71,72,73,74,75,76]. In LPS-activated RAW 264.7 cells, corilagin robustly inhibits the NF-κB pathway, restraining the release of TNF-α, IL-1β, IL-6, and COX-2 [31]. Moreover, in models of neuroinflammation and infection, it attenuates microglial activation and inflammatory injury of peripheral and central tissues by decreasing oxidative stress and pro-inflammatory mediator levels [31,77,78,79]. However, similarly to oenothein B, in the context of infection (*Leishmania major*-infected RAW 264.7 macrophages), Sanguiin H-6 has demonstrated preferential immune-stimulatory activity, boosting host defense mechanisms, most notably through the upregulation of cytokine expression. This context-dependent profile highlights its complex immunomodulatory activity: exerting anti-inflammatory actions under conditions of sterile inflammation, while enhancing immune response and antimicrobial defense under infectious conditions [80].

Geraniin-rich plant extracts have also consistently exhibited highly potent antioxidant activity, as evidenced across various model systems [17,81]. The ethanolic extract of *Nephelium lappaceum* peels demonstrated DPPH radical scavenging comparable to vitamin C, but with minimal pro-oxidant activity, paralleling commercial antioxidants [82]. Geraniin further displayed 5- to 6-fold greater FRAP activity than L-ascorbic acid and Trolox, efficiently neutralizing a spectrum of radicals, including DPPH, superoxide, hydroxyl, and nitric oxide, with IC_50_ values 7–14 times lower than conventional antioxidants such as ascorbic acid, BHA, and BHT [83,84,85]. Notably, geraniin mitigates cigarette smoke-induced oxidative stress in lung epithelial cells by reducing ROS accumulation and preserving cellular antioxidant defenses [86]. This exceptional capacity results from hydrogen atom transfer, generating resonance-stabilized phenoxyl radicals that terminate chain reactions. Geraniin also inhibits xanthine oxidase and lipid peroxidation, increases glutathione levels, and activates Nrf2 signaling, facilitating antioxidant gene expression [87,88,89]. In vivo studies reveal that geraniin mitigates liver injury in carbon tetrachloride-treated rats by enhancing hepatic glutathione and catalase, ameliorating oxidative damage, and improving liver function [17]. Its anti-inflammatory effects extend to inhibition of 5-lipoxygenase and iNOS in macrophages, suppression of NF-κB activation, and enhancement of macrophage phagocytosis [87,90].

Oenothein B, a dimeric macrocyclic ellagitannin prevalent in *Epilobium* and *Oenothera* species, is a well-characterized immunomodulator and antioxidant. Oenothein B-rich extracts are increasingly recognized for their potent anti-inflammatory, antioxidant, and tissue-protective properties. In a recent in vivo study by Simeonova et al., *E. angustifolium* extract significantly reduced colonic inflammation and oxidative stress in a DSS-induced ulcerative colitis mouse model, lowering myeloperoxidase and malondialdehyde levels, enhancing antioxidant enzyme activities, and improving hematological parameters with efficacy comparable to reference dexamethasone treatment [91]. Numerous independent in vivo studies confirm that oenothein B exerts protective effects in a variety of inflammatory models, including the amelioration of hepatic injury in alcoholic liver disease and intestinal remodeling in chemically induced ulcerative colitis [91,92,93,94]. These findings underscore the therapeutic relevance of oenothein B as a principal bioactive constituent in Epilobium-based preparations. The pharmacodynamic profile of this unique ellagitannin derivative includes reduced ROS production in stimulated neutrophils, inhibition of myeloperoxidase release with potency comparable to indomethacin, and attenuation of pro-inflammatory enzymes, including hyaluronidase and lipoxygenase [92,95]. In terms of immune responses, oenothein B activates phagocytes, induces intracellular calcium flux, ROS generation, NF-κB activation, and cytokine release in vitro [58,96]. In vivo, it triggers neutrophil infiltration and keratinocyte activation, and recent studies reveal significant neutrophil recruitment and induction of the chemokine KC [58,97,98]. Importantly, the intact dimeric macrocyclic structure is crucial for this activity, as smaller polyphenols lack similar immunostimulatory properties [58]. Further supporting its immunomodulatory profile, oenothein B dose-dependently suppresses iNOS gene and protein expression in LPS-stimulated macrophages, inhibits Toll-like receptor-mediated responses via NF-κB pathways, and stimulates γδ T and natural killer cells to produce IFN-γ, thereby enhancing anti-infective and antitumor immunity. It also regulates dendritic cell maturation and cytokine production through apoptosis-inducing mechanisms, establishing its role as a potent immune modulator with clear clinical and therapeutic relevance [97,99,100].

Chebulagic and chebulinic acids, principal ellagitannin components of *Terminalia chebula* extracts, demonstrate potent anti-inflammatory activity in both in vitro and in vivo models. Chebulagic acid attenuates LPS-induced inflammation in macrophage cell lines by downregulating iNOS and COX-2 expression, suppressing pro-inflammatory cytokines such as TNF-α and IL-6, and inhibiting nuclear translocation of NF-κB and STAT1/3 [101]. In a variety of animal models, chebulagic acid reduced edema, joint swelling, and inflammatory cell infiltration and mitigated symptoms in induced arthritis and colitis, supporting its therapeutic potential as an anti-inflammatory agent [13,102].

Chebulinic acid displays similar, albeit less pronounced effects. It has been demonstrated to efficiently inhibit nitric oxide production and COX-2 expression in activated macrophages, reduce pro-inflammatory chemokines and cytokines in vitro, and alleviate symptoms of acute inflammation and oxidative damage in mouse models [13,101,102]. Additionally, chebulinic acid acts as an anti-adhesive agent in bacterial inflammation, disrupting host–pathogen interactions, and provides protection against inflammatory bone loss, liver injury, and ulceration in various experimental settings [13].

The marked difference in anti-inflammatory potency between chebulagic acid and chebulinic acid is primarily attributable to their distinct structural features; specifically, chebulagic acid’s incorporation of a rigid hexahydroxydiphenoyl (HHDP) group in place of two galloyl moieties confers increased spatial hindrance and reduced flexibility, which enhances its interaction with biological targets and underlies its superior anti-inflammatory, antioxidant, and anti-infective effects compared to the more flexible, triple-galloyl chebulinic acid, as demonstrated in computational studies [59,103].

### 4.2. Antimicrobial Properties

The broad-spectrum antimicrobial activities of ellagitannins—including antibacterial, antiviral, and antiprotozoal effects, are well documented and continue to garner scientific interest as new drug-resistant pathogens emerge [26,104,105,106]. A compelling area of research concerns their synergistic potentiation of conventional antibiotics, which has been observed in the inhibition of key drug-resistant organisms such as methicillin-resistant Staphylococcus aureus (MRSA) and carbapenem-resistant Acinetobacter baumannii, among others [107]. Importantly, their antiviral properties are equally significant; chebulagic acid and chebulinic acid, among others, have demonstrated potent inhibition of viruses including, SARS-CoV-2, herpes simplex virus (HSV-2) and influenza A virus (IAV), acting through mechanisms such as blocking viral entry, attachment, and progeny release [13,59,108,109]. Such attributes emphasize the relevance of representative ellagitannins as promising lead compounds in the ongoing search for novel, adjunctive anti-infective therapies, targeting both resistant bacterial pathogens and clinically significant viral infections in the face of escalating global antibiotic and antiviral resistance.

Pomegranate has emerged as a promising natural agent for the prevention and management of diverse respiratory disorders. Punicalagin, the principal ellagitannin in various parts of Punica granatum, demonstrates broad-spectrum antimicrobial activity against Gram-positive and Gram-negative bacteria, fungi, and significant respiratory viruses. Its mechanisms include disruption of microbial cell walls, inhibition of biofilm formation, modulation of quorum sensing, and interference with essential microbial enzymes. Notably, punicalagin is effective against antibiotic-resistant strains and clinically relevant respiratory and urinary pathogens, including gentamicin-resistant Pseudomonas aeruginosa, Staphylococcus aureus, Escherichia coli, and Proteus mirabilis, supporting its role in addressing antimicrobial resistance [11,110].

In the context of viral infections, punicalagin exhibits virucidal activity against influenza viruses (A, H1N1, H3N2, B), herpesviruses, RSV, and SARS-CoV-2, acting through inhibition of viral entry, RNA replication, and progeny release [10,111,112,113]. Notably, ellagitanin-rich pomegranate extracts synergistically enhance the antiviral efficacy of oseltamivir, suppressing the replication of the influenza A virus [114]. More importantly, the ellagitannin has been found to act as an allosteric inhibitor of SARS-CoV-2 main protease (3CLpro), essential for viral polyprotein processing, and disrupt the interaction between the viral spike protein and the ACE2 receptor, potentially blocking viral entry [115]. Another recent study identified punicalagin as a potent inhibitor of the SARS-CoV-2 NSP13 helicase, a crucial enzyme for viral replication, demonstrating direct binding to the enzyme (KD = 21.6 nM), inhibiting ATP hydrolysis and DNA binding, and subsequently suppressing viral replication in A549-ACE2 and Vero cells with EC_50_ values of 347 nM and 196 nM, respectively [113]. According to the most recent findings, punicalagin potently inhibits the formation of liquid condensates between the nucleocapsid protein and viral RNA—a process essential for replication of multiple viruses, resulting in broad-spectrum antiviral activity. At nanomolar concentrations, it reduced tissue viral load and mitigated virus-induced inflammation and lethality in mice infected with SARS-CoV-2, vesicular stomatitis virus, and influenza A virus, by specifically targeting MAVS-dependent inflammatory pathways [116]. Punicalagin further attenuated inflammation-associated signaling in respiratory viral infections, alleviating airway inflammation and conferring protection in experimental models [117]. These multi-targeted activities are accompanied by a favorable safety profile, making punicalagin-rich extracts attractive candidates for adjunctive therapy, particularly in high-risk populations vulnerable to severe COVID-19, including the elderly and individuals with comorbidities like diabetes and cardiovascular disease [112].

Analogous to punicalagin’s antiviral effects, sanguiin H-6 (SH6) and related ellagitannins from *Rubus* species also exhibit significant antimicrobial and antiviral activities. Of particular note, recent in silico and in vitro studies have demonstrated that SH6 strongly binds to key SARS-CoV-2 targets, including the S1 and S2 subunits of the spike protein and the main protease (Mpro), with superior binding affinities compared to other polyphenols, inhibiting both viral entry and replication [52]. SH6′s potent antibacterial activity spans numerous Gram-positive and Gram-negative species, with minimum inhibitory concentrations reported for *Streptococcus A*, *Streptococcus pneumoniae*, *Corynebacterium diphtheriae*, *Bacillus subtilis*, *Clostridium sporogenes*, *Staphylococcus aureus*, *Staphylococcus epidermidis*, and *Moraxella catarrhalis* [118]. It also significantly inhibits *Escherichia coli* and *Clostridium perfringens*, and exerts fungistatic activity against *Candida albicans* [104,119].

Corilagin has demonstrated inhibitory effects against a broad spectrum of microorganisms, including bacteria, fungi, and viruses. While its activity against many pathogens such as *Escherichia coli*, *Pseudomonas aeruginosa*, *Klebsiella pneumoniae*, and *Bacillus subtilis* is generally weaker, corilagin exhibits potent antibacterial effects against *Staphylococcus aureus*, with reported minimum inhibitory concentrations (MICs) ranging from 25 to 256 μg/mL [120,121]. Importantly, corilagin has shown the ability to significantly reduce the MICs of β-lactam antibiotics against both β-lactamase-positive and -negative methicillin-resistant Staphylococcus aureus (MRSA) strains by inhibiting penicillin-binding protein 2′ (PBP2a), a key factor in MRSA antibiotic resistance [122,123,124]. Additionally, the ellagotannin displays strong antifungal activity, notably against *Candida glabrata* (MIC = 0.5 μg/mL) and inhibits chitin synthase II in Saccharomyces cerevisiae (IC50 = 25 μg/mL) [30,125]. It also effectively inhibits Helicobacter pylori (MIC = 4 μg/mL) and demonstrates substantial activity against multidrug-resistant *Acinetobacter baumannii* [126,127]. Corilagin’s broad-spectrum antiviral properties are equally noteworthy, as it inhibits key viral enzymes and entry processes for pathogens such as Epstein–Barr virus, HSV-2, hepatitis C virus, HIV-1, and notably SARS-CoV-2, where it blocks the interaction between the viral spike protein and ACE2 receptor, disrupts viral entry, and suppresses replication without cytotoxic effects, supporting its development as a multifaceted antiviral agent [128,129].

Geraniin-rich plant extracts exhibit notable antimicrobial effects against major human pathogens such as *S. aureus*, *P. aeruginosa*, *S. pyogenes*, *and E. coli* [130], and display strong antiviral activity, including the inhibition of dengue virus type 2 and enterovirus 71 by interfering with viral attachment and replication [131,132,133]. It has also been found to exert potent dose-dependent inhibition of viral entry and HIV-1 reverse transcriptase, impeding HIV infection by blocking gp120-CD4 interactions [134]. Additional studies confirm suppression of hepatitis B virus antigen release and inhibition of retroviral enzymes (protease, integrase, reverse transcriptase) [135,136,137]. In vivo, geraniin improves survival and reduces disease in EV71-infected mice, with evidence for dose-dependent efficacy [131]. Beyond antibacterial and antiviral effects, geraniin also exhibits antifungal activity against *Candida* species, and inhibits *Plasmodium falciparum* and Leishmania donovani, supporting its potential as a broad-spectrum agent against fungal and parasitic infections [138,139].

Ellagitannins, such as oenothein B have also demonstrated potent synergistic effects with β-lactam antibiotics against MRSA, reducing minimum inhibitory concentrations by several hundred-fold and potentially restoring antibiotic susceptibility [60]. In vitro studies on *E. angustifolium* and *E. parviflorum* extracts rich in oenothein B have shown strong antibacterial activity (MICs 4–512 µg/mL) against Gram-positive and Gram-negative prostatitis-causing pathogens, with efficacy comparable to levofloxacin [23]. The study also highlighted the importance of purification processes to enrich bioactive constituents, particularly oenothein B, thereby enhancing the antibacterial efficacy of the extracts. Ellagitannin-rich extracts from *Epilobium hirsutum* and *Chamerion angustifolium* (Onagraceae) effectively reduce HIV-1 release and selectively trigger apoptosis in HIV-1-infected T cells, sparing healthy cells. This activity was found to be driven by a synergistic interplay between oenothein B and quercetin, which jointly block viral replication and promote the elimination of infected cells. Moreover, extracts from *C. angustifolium* and *Senna alexandrina* further diminish the infectivity of progeny virions, supporting their utility as adjunctive agents targeting late stages of the HIV-1 lifecycle [140].

Chebulinic and chebulagic acids are also potent antiviral agents that inhibit the neuraminidase-mediated release of influenza A virus (IAV), maintaining efficacy even against oseltamivir-resistant NA/H274Y mutants, thus representing promising leads for novel neuraminidase inhibitors [59]. Both compounds also demonstrate significant antibacterial activities: *Sapium baccatum*-derived chebulinic acid acts against *Ralstonia solanacearum* and multidrug-resistant *Acinetobacter baumannii*, while *Terminalia macroptera* leaf extracts show strong effects against *N. gonorrhoeae*, including resistant strains [126,141]. Chebulinic acid also functions as a potent DNA gyrase inhibitor, effective against *Mycobacterium tuberculosis*, including quinolone-resistant strains, with molecular modeling confirming stable inhibition of the enzyme [142]. Similarly to punicalagin, it also exhibits anti-HIV activity, blocking gp120-CD4 binding to prevent viral entry and, along with chebulagic acid, inhibits HSV-2 infection by impairing viral attachment and penetration in a potent dose-dependent manner (IC50: 0.06–1.41 μg/mL) [109,143]. Finally, aqueous extracts of *Terminalia chebula* containing chebulinic acid reveal broad antifungal activity, particularly against dermatophytes and *Candida* spp. [144].

### 4.3. Therapeutic Potential in Cardiovascular Diseases (CVDs)

Ellagitannins have gained attention for their potential role in CVDs prevention and therapy. Increasing evidence from preclinical and clinical studies demonstrates that ellagitannins and their gut-derived metabolites, particularly urolithins, exert anti-inflammatory, anti-atherosclerotic, and endothelial-protective effects, improving cardiovascular risk factors such as lipid profile, vascular function, and oxidative stress. These emerging findings position ellagitannins as promising candidates for adjunctive dietary-based strategies to reduce the global burden of CVDs.

Punicalagin confers significant cardioprotective effects through multiple mechanisms: it enhances lipid metabolism, reduces LDL oxidation, and improves endothelial function, while its potent antioxidant and anti-inflammatory properties slow atherosclerosis progression [50,145]. Studies demonstrate that pomegranate preparations rich in punicalagin improve lipid profiles in diabetes models, reduce cardiac fibrosis, and restore cardiac metabolic function by activating AMPK pathways, promoting mitochondrial health, and reducing oxidative stress [146,147]. Experimental models reveal that punicalagin pretreatment mitigates cardiac trauma during ischemia/reperfusion injury, decreases infarct size, lowers CK-MB and LDH levels, biomarkers of myocardial damage, and prevents apoptosis, effects attributed to AMPK activation and ACC phosphorylation [148]. Beyond maintaining antioxidant capacity and improving troponin T, it mitigates heart tissue apoptosis, inflammation, and DNA damage by regulating the apoptotic machinery, including caspase activity, p53, and the BCL-2 pathways [149]. Notably, clinical trials demonstrate that hydroxytyrosol and punicalagin supplementation results in reductions in oxidized LDL and improvements in vascular function and blood pressure among middle-aged adults with elevated cardiovascular risk. These findings support its potential as a cardioprotective nutraceutical, although solubility and bioavailability issues necessitate further clinical trials to establish optimal dosing regimens and delivery methods [50,150,151].

The cardioprotective effects of corilagin have also been studied in preclinical models of Ang II-induced atrial fibrosis and atrial fibrillation. By modulating the PI3K/Akt and NF-κB pathways, corilagin attenuated fibrotic tissue changes, reduced oxidative stress, and suppressed Ang II-induced upregulation of fibrotic markers, including smooth muscle alpha-actin (α-SMA), connective tissue growth factor (CTGF), collagen I, and collagen III, significantly reducing arrhythmogenic risk [152]. The ellagitannin additionally lowers blood pressure in hypertensive rats, likely through reduced noradrenaline release and vasorelaxation [153]. It also demonstrated thrombolytic potential by modulating plasminogen activators and inhibited atherosclerosis progression by reducing oxidative damage and vascular smooth muscle cell proliferation [154]. Moreover, it attenuated doxorubicin-induced cardiotoxicity via modulation of PI3K/Akt and NF-κB signaling pathways in a rat model [155].

Being a precursor to corilagin, geraniin has also demonstrated significant antihypertensive effects in vivo, primarily through inhibition of angiotensin-converting enzyme (ACE), leading to vasodilation and blood pressure reduction [84]. It is also reported to modulate sympathetic activity by decreasing noradrenaline release at peripheral nerve terminals, independent of adrenal function. It is reported to decrease sympathetic neurotransmission by restraining noradrenaline liberation at peripheral nerve terminals [153]. Additionally, geraniin exhibits antithrombotic activity by inhibiting platelet aggregation in response to different activating stimuli and disrupting platelet-neutrophil interactions critical for thrombus formation. It also enhances fibrinolysis by inhibiting plasminogen activator inhibitor-1 (PAI-1), promoting clot breakdown, and offering potential benefits in thromboembolic and atherosclerotic conditions [17].

Similarly, chebuloyl-type ellagitannins have also been demonstrated to exert broad cardioprotective effects. Bag et al. reported that pre-administering *T. chebula* extract to rats reduced lipid peroxide formation and normalized marker enzyme activities in a model of isoproterenol-induced toxicity [156]. In a rat model, chebulinic acid effectively inhibited aortic contractile responses to serotonin and angiotensin II and competitively blocked prazosin binding to vascular receptors in a concentration-dependent manner. Binding assays reveal reversible, non-specific inhibition of adrenergic receptors, reducing receptor affinity and maximal binding sites, which collectively contribute to its vasodilatating properties. It also reversibly reduces maximal left ventricular pressure in isolated rat hearts at nanomolar concentrations, indicating its ability to modulate both vascular and cardiac contractility [157].

### 4.4. Therapeutic Potential in Metabolic Disorders

Metabolic disorders, such as diabetes, obesity, dyslipidemia, and metabolic syndrome, are tightly linked to the development and progression of cardiovascular diseases (CVDs), acting as key drivers of vascular dysfunction, atherosclerosis, and cardiac complications. The management and prevention of metabolic pathologies are therefore vital for reducing CVD morbidity and mortality. Both prior and recent studies suggest that ellagitannins and their gut-derived metabolites may positively influence metabolic pathways linked to both cardiovascular and metabolic disorders, offering a promising approach for the prevention and management of these overlapping conditions.

The antioxidant and anti-lipid peroxidation capacity of punicalagin, demonstrated in vitro, contributes to its ability to mitigate obesity-related oxidative stress and inflammation through activation of the Nrf2/Keap1 signaling axis and modulation of macrophage polarization [158]. Consistent with these findings, in vivo studies demonstrate that punicalagin supplementation regulates lipid homeostasis, enhances mitochondrial function, attenuates non-alcoholic fatty liver disease, strengthens pancreatic β-cell function and improves glycemic control in animal models [159,160]. It further supports metabolic health by inhibiting carbohydrate-digesting enzymes (α-amylase and α-glucosidase), enhancing insulin sensitivity, and reducing oxidative stress associated with hyperglycemia [9]. The ellagitannin has also demonstrated promising renoprotective effects in diabetes, cisplatin-induced acute kidney injury, and lupus nephritis models, supporting its potential as a therapeutic agent for diabetes-related complications [49,161]. In diabetic nephropathy, which affects up to 50% of end-stage renal disease patients, punicalagin showed therapeutic potential by downregulating NOX4 and inhibiting pyroptosis via the TXNIP/NLRP3 inflammasome pathway [162].

Corilagin’s antidiabetic effects are also well-documented. Functioning as a potent in vitro inhibitor of intestinal α-amylase and α-glucosidase, it effectively limits carbohydrate breakdown and glucose absorption. In animal studies, corilagin has been shown to improve insulin sensitivity by activating the PPARγ pathway, mirroring the mechanism of established antidiabetic medications such as rosiglitazone [163,164,165]. In diabetic rodent models, similar beneficial effects on glucose homeostasis have also been reported for geraniin and geraniin-rich extracts, restoring pancreatic β-cell function, enhancing insulin secretion, and improving insulin sensitivity in liver, adipose tissue, and skeletal muscles. Additionally, geraniin may alleviate serious diabetic complications such as nephropathy, retinopathy, encephalopathy, and cardiomyopathy. Many of its metabolites, including corilagin, ellagic acid, and urolithins, also exhibit anti-advanced glycation end-product (AGE) activity, reducing oxidative stress, inflammation, and fibrosis in affected organs [89].

Notable anti-obesity and anti-diabetic actions have also been demonstrated by macrocyclic ellagitannins, such as oenothein B. In a Caco-2 model, it has been shown to decrease fructose uptake by the intestinal GLUT5 transporter, which may help reduce de novo lipogenesis in the liver [166]. In vivo studies show that oenothein B supplementation reduces visceral fat, improves inflammatory markers associated with metabolic dysfunction, and boosts antioxidant defenses, including superoxide dismutase, catalase, and glutathione [92,166]. Experimental results in *C. elegans* indicate that oenothein B inhibits fat accumulation by modulating the expression of genes linked to lipid synthesis, oxidation, and storage (e.g., mdt-15, fat-5, and fasn-1), and upregulating key antioxidant enzymes, such as SOD and GSH-Px [167]. Although direct clinical trials for oenothein B remain limited, its safety profile and efficacy in preclinical models warrant further research as a well-tolerated adjuvant for managing metabolic disorders.

The metabolic attenuation and antidiabetic profile of chebulinic acid (*Terminalia chebula*), are also multifaceted and supported by both in vitro and animal studies. The ellagitannin has been shown to act as a dual inhibitor of protein tyrosine phosphatases PTPN9 and PTPN11 and promote glucose uptake through activation of the AMPK pathway [168,169]. Further evidence from studies on *T. chebula* extracts demonstrates α-glucosidase inhibition and improved insulin receptor signaling, which leads to upregulated GLUT4 translocation and better glucose handling in muscle tissue [170]. Additional benefits of chebulinic acids include suppression of aldose reductase activity and advanced glycation end product formation, key contributors to diabetic complications [171].

### 4.5. Anticancer Activity

Cancer ranks as the second leading cause of death globally after cardiovascular disease. Recent research highlights ellagitannins’ capacity to inhibit carcinogenesis and tumor progression through targeting multiple hallmarks of neoplastic diseases, affecting cell survival, apoptosis, invasion, and angiogenesis. Their key metabolites, namely ellagic acid and urolithins, execute these effects by reducing oxidative stress, modulating cellular signaling networks involved in tumor survival (PI3K/Akt, MAPK/ERK, Wnt/β-catenin and NF-κB), as well as direct regulation of apoptosis-related proteins (Bcl-XL, Bax, caspases, PARP) and EMT/migration markers (GOLPH3, MMP-2, MMP-9, N-cadherin, E-cadherin, TIMPs, Snail, Slug). Evidence from cell and animal studies further supports their ability to induce cell cycle arrest and apoptosis and suppress angiogenesis, pointing to their potential use as adjunctive, low-toxicity agents in cancer prevention and therapy.

Pomegranates, rich in ellagitannins, show considerable promise in cancer prevention and management [172]. Punicalagin’s anticancer activity is partially attributed to its capacity to selectively intensify oxidative stress, disrupt mitochondrial function, and promote senescence and apoptosis in malignant cells, while sparing normal cells. In breast cancer models, punicalagin substantially diminished the metastatic potential of MDA-MB-231 cells and reversed epithelial–mesenchymal transition (EMT) by downregulating GOLPH3, MMP-9, MMP-2, and N-cadherin levels [173]. Similarly, in prostate cancer models, the ellagitannin demonstrated significant inhibition of cell proliferation and induction of apoptosis with minimal toxicity to normal cells [174].

The anticancer activity of both punicalagin and pomegranate juice has also been studied in various colorectal cancer models. In HT-29 and HCT116 cells, treatment was found to effectively inhibit critical pro-inflammatory and survival pathways, including TNF-α-induced COX-2 expression, NF-κB nucleus translocation, AKT activity, and Anx-A1 expression, resulting in reduced proliferation and enhanced cell death [9,175,176]. Cervical cancer cells, including HeLa, exhibited dose-dependent reductions in proliferation and migration upon punicalagin treatment, mediated by multifaceted mechanisms. Besides suppressing NF-κB signaling, it downregulated matrix metalloproteinases (MMP-2, MMP-9), β-catenin, and the anti-apoptotic Bcl-2 protein, while upregulating the pro-apoptotic factors Bax and tissue inhibitors of metalloproteinases (TIMP-2, TIMP-3). In addition, punicalagin induced G1 cell cycle arrest and triggered mitochondrial apoptosis in malignant cells [177,178]. Lung carcinoma models also display marked sensitivity to punicalagin-induced cell death at low micromolar concentrations, related to a preferential G1/S arrest of the cell cycle and caspase activation in malignant populations [179]. In glioma, particularly the U87MG cell line, the ellagitannin activates both apoptotic and autophagic cytotoxic pathways. Thyroid cancers, including papillary, anaplastic, follicular, and medullary subtypes, respond to punicalagin via ATM-mediated DNA damage signaling and induction of senescence, as indicated by SA-β-gal staining and altered morphology. The anti-metastatic and tumor-suppressive actions of punicalagin have been further validated in vivo, with rat thyroid cancer models showing pronounced inhibition of tumor growth and metastatic progression [180,181].

Sanguiins have also garnered significant attention for their potential anticancer properties. Sanguiin H-6 has been shown to exert antineoplastic effects primarily through inhibition of DNA topoisomerases I and II. It demonstrated cytostatic activity against HeLa cells, suppressing proliferation at an effective concentration of 12 µM, with a dose-dependent reduction in topoisomerase activity. Additionally, SH6 exhibited notable antiangiogenic effects [53]. In studies involving HT1080 human fibrosarcoma cells, SH6 inhibited the binding of KDR/Flk-1-Fc to VEGF165 in a concentration-dependent manner and suppressed VEGF-induced proliferation of human umbilical vein endothelial cells (HUVECs) [182]. It has also been found to reduce metastatic potential of A549 lung cancer cells by modulating the TGF-β1-induced Smad 2/3 signaling pathway, enhancing epithelial E-cadherin expression and repressing mesenchymal markers Snail and N-cadherin during epithelial–mesenchymal transition [51]. In breast cancer models, SH6 exhibited antiproliferative cytotoxic effects in both tripple negative (MDA-MB-231) and adriamycin-resistant MCF-7/Adr cells and reduced the metastatic activity of MDA-MB-231 cells by downregulating VEGF, phosphorylated Akt, and ERK1/2 signaling pathways [183,184]. Moreover, SH6 increased the pro-apoptotic Bax/Bcl-2 ratio in both breast cancer models, shifting the fate of cancer cells toward programmed cell death. In A2780 human ovarian carcinoma cells, SH6 induced antiproliferative effects and morphological changes consistent with apoptosis without causing cell cycle arrest. It triggered early apoptotic events, including caspase activation, PARP cleavage, and activation of the MAPK signaling pathway. Increased levels of the truncated p15/BID proapoptotic protein were also reported, triggering cytochrome c release and activation of intrinsic apoptosis pathways [183,185,186].

Corilagin also displays strong antiproliferative activity across a wide range of human and murine tumor models, effectively inhibiting tumor growth and prolonging survival through a multifaceted array of molecular mechanisms. As a broadly acting antitumor agent, it consistently inhibited the proliferation and enhanced cellular sensitivity to cytotoxic agents of numerous human cancer cell lines, including nasopharyngeal, pancreatic (Bxpc-3), hepatocellular (Hep3B, SMMC7721, Bel7402, H22), laryngeal (Hep-2), gastric (SGC-7901), ovarian (SKOv3ip, Hey), lung adenocarcinoma, colon (HCT-8), glioblastoma multiforme (GBM), osteosarcoma (OS-732), metastatic carcinoma and cholangiocarcinoma [30]. It demonstrated potent antitumor activity against ovarian cancer by selectively inhibiting cancer cell growth and displaying minimal toxicity toward normal cells. It suppressed the expression of key cell cycle regulators such as Cyclin B1, Myt1, Phospho-cdc2, and Phospho-Weel, inducing cell cycle arrest at the G2/M phase and apoptosis signaling. In vivo, corilagin significantly reduces xenograft tumor growth and uniquely blocks TGF-β secretion and Snail stabilization, effectively inhibiting both canonical Smad and non-canonical ERK/AKT pathways [187]. The administration of corilagin markedly reduced tumor burden in models of Hep3B hepatocellular carcinoma, fibrosarcoma (sarcoma-180), Lewis lung carcinoma, lymphocytic leukemia (P388), hepatic carcinoma (H22), osteosarcoma, and colon carcinoma, resulting in significant prolongation of the overal survival across xenograft models [188,189]. Importantly, the synergistic interaction of corilagin with established chemotherapeutic drugs, such as cisplatin, doxorubicin, and temozolomide, has been shown to enhance chemosensitivity in hepatocellular carcinoma and glioma models, supporting its role in overcoming drug resistance [187,190].

The anticancer activity of geraniin is also validated in a broad range of human and murine malignancies, including colorectal, breast, melanoma, bladder, and lung cancer models, as well as Jurkat and HeLa carcinoma cell lines. In vitro, the ellagitannin reduces cell viability and proliferation in a dose- and time-dependent manner, inducing cell cycle arrest and triggering apoptosis through activation of the Fas death receptor and the downstream extrinsic pathway of apoptosis [20]. Additional mechanisms in orchestrating programmed cell death include targeted degradation of focal adhesion kinase, promotion of DNA fragmentation, and suppression of DNA repair processes. Moreover, geraniin disrupts oncoprotein stability through direct inhibition of both Hsp90 chaperone and ATPase functions, leading to profound decreases in Raf-1, phosphorylated Akt, and EGFR protein levels. These combined effects closely mirror the molecular actions of classical Hsp90 inhibitors, resulting in the abrogation of key prosurvival and proliferative signaling pathways within malignant cells [191]. Notably, in aggressive breast cancer 4T1 xenografts, daily geraniin administration attenuates primary tumor growth and hepatic metastasis and alters plasma protein expression profiles related to tumor aggressiveness [192]. In colorectal cancer, geraniin was shown to inhibit proliferation and induce catastrophic chromosomal instability, further promoting cytostasis and apoptosis, while selectively sparing normal mucosal cells [193,194]. Similarly to its active metabolite corilagin, geraniin has the capacity to effectively reverse TGF-β1-driven epithelial–mesenchymal transition and restore sensitivity to anoikis, resulting in disruption of the metastatic phenotype [195].

The antineoplastic activity of the ellagitannin family is also shared by macrocyclic representatives, including oenothein B. It has demonstrated strong antiproliferative and proapoptotic effects in a wide array of cancer models, including oral, cervical, prostate, hepatocellular, leukemia, and non-small cell lung cancer cell lines, while exhibiting lower cytotoxicity toward non-malignant cells compared to conventional chemotherapeutics [60,75,196,197]. In A549 lung adenocarcinoma cells, oenothein B induced G1 phase cell cycle arrest and activated mitochondria-mediated apoptosis, characterized by elevated intracellular ROS and increased expression of pro-apoptotic markers, including cleaved caspase-3, PARP, cytochrome c, and Bax. These effects appeared to be strongly dependent on ROS-driven downregulation of the PI3K/Akt/NF-κB pathway, as evidenced by the ability of ROS scavenger NAC and PI3K agonist IGF-1 to reverse both apoptosis and pathway inhibition induced by oenothein B. These findings point to oxidative stress serving as a critical upstream regulator of cell cycle arrest and intrinsic apoptotic signaling in the NSCLC model [75].

In prostate cancer studies, oenothein B and related compounds inhibit tumor cell growth and survival both in cell cultures and animal models, trigger mitochondria-driven apoptosis, and effectively inhibit enzymes critical to disease progression (e.g., 5α-reductase, aromatase, and neutral endopeptidase), validating the traditional application of Epilobium extracts for prostate health [198,199,200].

Further mechanistic insight reveals that oligomeric ellagitannins, including oenothein B, act as potent inhibitors of poly(ADP-ribose) glycohydrolase (PARG), regulating gene expression and apoptotic cell death [201]. Importantly, macrocyclic ellagitannins also function as inhibitors of viral DNA polymerases, as demonstrated in studies with Epstein–Barr virus (EBV). EBV, a member of the herpesvirus family, is strongly implicated in the pathogenesis of several human cancers, such as nasopharyngeal carcinoma, lymphomas, and certain gastric cancers. This oncogenic potential is mediated through the virus’s ability to disrupt normal cellular processes, including promoting uncontrolled cell proliferation, suppressing apoptosis, and remodeling the tumor microenvironment via oncogenic signaling networks [202].

Chebulinic acid functions as a natural multi-target antitumor agent across numerous cancer models, demonstrating particularly strong efficacy in human colorectal carcinoma. As a principal bioactive monomer within the Triphala polyherbal formulation, the *T. chebula* ellagitannin is essential for Triphala’s potent anticancer activity. [203].

Numerous independent studies report on the anti-proliferative, pro-apoptotic, and anti-migratory effects of chebulinic acid in this particular model, which are mechanistically linked to suppression of the PI3K/AKT and MAPK/ERK pathways, activation of caspase-3, DNA fragmentation, PARP cleavage, and mitochondrial membrane permeabilization [203,204,205]. Chebulinic acid also prevents VEGF-driven angiogenesis by inhibiting VEGFR-2 phosphorylation, supporting anti-angiogenic benefit in solid tumor malignancies where neovascularization is pivotal [206].

Furthermore, chebulinic acid has been shown to induce apoptosis in human leukemia cell lines HL-60 and NB4, and to suppress proliferation in cervical carcinoma (HeLa) cells. Uniquely, it selectively inhibits mitochondrial AAC2 (adenine nucleotide translocase) carrier activity, disrupting ATP/ADP exchange and imposing energy stress on cancer cells [205,207]. In erythroid and vascular models, it regulates transcriptional activity, inhibits PDGF-BB-induced MMP-2 expression, and blocks Akt/ERK signaling to suppress vascular remodeling and neovascularization [208].

### 4.6. Neuroprotective Properties

Although extensive evidence supports the neuroprotective potential of ellagitannins, it is primarily their microbial metabolites, urolithins, that can reach sufficient concentrations in brain tissue, given the limited permeability of parent ellagitannins through the blood–brain barrier (BBB) [209]. The neuroprotective properties of ellagitannins and their metabolites are intricately linked to the modulation of neuroinflammatory and oxidative stress pathways central to neurodegenerative disease pathology. They have also been reported to mitigate neuronal damage by suppressing pro-inflammatory cytokines, inhibiting NF-κB and ERK1/2 signaling, and attenuating the activation of microglia and astrocytes—key mediators of neuroinflammation in Alzheimer’s disease, traumatic brain injury, and other CNS disorders. Moreover, ellagitannins target amyloid-β toxicity, mitochondrial dysfunction, and blood–brain barrier integrity, addressing both inflammatory and pathology-specific mechanisms implicated in the progression of neurodegenerative conditions.

Punicalagin’s neuroprotective potential has been substantiated in both neuronal cell cultures and animal models, with experimental evidence supporting its capacity to inhibit amyloid-beta aggregation and microglial activation, preserve cognitive function, and reduce neuronal death through antioxidant and anti-inflammatory mechanisms. In Alzheimer’s transgenic mouse models and studies involving neonatal brain hypoxia, pomegranate-derived ellagitannins mitigate lipid oxidation and prevent neuronal death. The dietary enrichment in pregnant mothers confers a significant reduction in neonatal brain tissue loss and apoptosis via downregulation of caspase-3 activity. Furthermore, punicalagin enhances methionine-sulfoxide reductase activity and mitochondrial aldehyde dehydrogenase, regulates Bcl-xL protein stability, activates antiapoptotic pathways, and reduces neuroinflammation by suppressing TRAF-6 and NF-κB signaling [63,210,211,212].

In models of Alzheimer’s and Parkinson’s diseases, corilagin has been found to inhibit amyloidogenic enzymes prolyl endopeptidase and β-secretase, modulate TLR4/Src/NOX2-driven ferroptotic pathways, and suppress oxidative stress, neuroinflammation, and iron accumulation in substantia nigra neurons, as confirmed in MPTP-induced mice and MPP+-treated N2a cells [213]. This multifaceted modulation of apoptotic, mitochondrial, and inflammatory pathways validates its promising application as a disease-modifying agent for neurodegenerative disorders.

Recent studies demonstrate that geraniin confers neuroprotection in both acute and chronic CNS injury models, further elucidating its mechanistic targets. In cerebral ischemia/reperfusion models, geraniin administration significantly reduces infarct volume, improves neurological deficits, and ameliorates neuronal pathology; these effects correlate with increased superoxide dismutase activity, decreased lipid peroxidation, and upregulation of the Nrf2/HO-1 signaling—hallmarks of antioxidant defense mechanisms in cytoprotection [214]. Moreover, rat models of spinal cord injury demonstrate that geraniin robustly attenuates oxidative stress, suppresses apoptotic activity, and mitigates neuroinflammation by restoring serum and tissue levels of the pro-inflammatory factors TNF-α, IL-1, IL-6 and COX-2 [215]. Geraniin administration also reversed haloperidol-induced neuronal damage in the striatum and ameliorated orofacial dyskinesia in rats [216]. Further experimental reports verify geraniin’s inhibitory activity on prolyl endopeptidase and β-secretase, facilitating preservation of neuropeptides and reduction in amyloid-β aggregation, implicated in Alzheimer’s pathogenesis [217,218].

Oenothein B confers neuroprotection in systemic inflammation models by dose-dependently suppressing pro-inflammatory cytokines IL-1β and IL-6, improving behavioral deficits, and reducing microglial activation and COX-2 expression in limbic brain regions. Due to its large, hydrophilic structure and limited ability to cross the BBB, it is likely that its neuroprotective effects are mediated indirectly through peripheral anti-inflammatory actions, or possibly via microbiota-derived metabolites that influence central neuroinflammatory responses [219,220].

Chebulagic and chebulinic acids, principal hydrolyzable tannins in *Terminalia chebula* and related species, exhibit substantial neuroprotective activity through multiple, complementary mechanisms. Chebulinic acid has shown significant efficacy against glutamate-induced excitotoxicity, a process characterized by excessive glutamate receptor activation, uncontrolled calcium influx, mitochondrial dysfunction, and ultimately neuronal death. Neuronal protection is mediated through inhibition of the MAPK pathway, restriction of calcium overload, and mitigation of oxidative stress. Additionally, chebulinic acid promotes neuronal survival by modulating the balance between pro- and anti-apoptotic factors and shifting the Bcl-2/Bax ratio [221]. Importantly, synergistic interactions with Boeravinone B further enhance antioxidant defenses and restore critical regulators of cellular redox homeostasis and aging [222]. The structurally related ellagitannin, chebulagic acid, extends these benefits by activating autophagy in SH-SY5Y human neuroblastoma cells. By engaging the AMPK–mTOR–Beclin-1–LC3 signaling axis, chebulagic acid facilitates the removal of dysfunctional proteins and rescues neurons in an MPP model of Parkinson’s disease [223]. Both chebulagic and chebulinic acids inhibit acetylcholinesterase, boost cholinergic activity, counter amyloid β aggregation, and reduce β-amyloid-driven toxicity, providing further mechanistic basis for their use in cognitive impairment and Alzheimer’s disease [12]. Recent reviews and in vivo findings further corroborate *T. chebula* extracts’ ability to reduce neuroinflammation, restore synaptic plasticity, improve memory and cognitive function, and confer safety in both animal and cellular models, consolidating the therapeutic potential of the plant in the prevention and potential treatment of neurodegenerative disorders [12,224].

### 4.7. Anti-Osteoporotic Effects

A growing body of evidence highlights the bone-protective efficacy of ellagitannin derivatives through complementary mechanisms targeting both osteoclastogenesis and inflammatory bone loss.

Punicalagin exhibits significant bone-sparing properties by inhibiting osteoclastogenesis and bone resorption in both in vitro and in vivo models. Mechanistically, punicalagin suppresses the differentiation of osteoclast precursor cells through downregulation of key osteoclast-specific marker genes, such as NFATc1, cathepsin K, c-Src, and TRAP. Its modulatory activity is also mediated through the targeted suppression of RANKL-induced activation of NF-κB and MAPK signaling cascades, which are fundamental for osteoclast maturation, survival, and bone-resorptive activity [225]. In animal models of estrogen deficiency (ovariectomized rodents) and diabetes, punicalagin administration not only preserved trabecular architecture and bone mineral density, but also attenuated systemic and local inflammation through suppression of pro-inflammatory cytokine production. Recent investigations further suggest that punicalagin may interfere with osteoclast actin-ring formation, macrophage fusion and multinucleation, critical processes in bone resorption, contributing to its therapeutic relevance in the treatment of osteoporosis and other bone metabolic disorders [225,226].

The in vitro anti-osteoporotic activity of sanguiin H-6 was demonstrated by its capacity to suppress osteoclast differentiation and bone loss, reduce reactive oxygen species production, and prevent the nuclear translocation of key transcription factors, including nuclear factor of activated T cells cytoplasmic-1 (NFATc1), c-Fos, and nuclear factor-κB (NF-κB). These mechanistic effects have also been validated in vivo, where ellagitannin supplementation inhibited osteoclast differentiation and prevented pathological bone loss [227]. Moreover, both sanguiin H-6 and *R. coreanus* extract have been shown to exhibit notable estrogenic activity, as evidenced by ERα-dependent proliferation in MCF-7 cells, suggesting potential benefits in postmenopausal osteoporosis [228].

Similarly to punicalagin, geraniin and its active metabolite corilagin have been found to suppress osteoclast differentiation and maturation and ameliorates osteolysis in mouse model by modulating RANKL-induced signaling pathways [229,230]. In ovariectomized rat models of postmenopausal osteoporosis, geraniin produced a marked increase in bone mineral density, trabecular bone volume, and overall bone microarchitecture, accompanied by improvement in key bone turnover (β-CTx, osteocalcin, ALP, BGP) and inflammatory (TNF-α, IL-1, and IL-6) markers in local and peripheral tissues. These effects are synergistically enhanced by the activation of the Wnt/β-catenin pathway and regulation of osteoblastic proliferation and differentiation, as well as upregulation of osteogenic transcription factors Runx2 and osterix [231].

Comparable pathways in punicalagin’s, corilagin’s, and geraniin’s antiosteoporotic activities suggest a shared structural and mechanistic basis for osteoprotection, strengthening the concept that their common metabolic products (i.e., urolithins A and B) act as the principal mediators of ellagitannins’ biological effects.

Reported biological activities, structural and mechanistic features of the representative ellagitannin derivatives are summarized in Table 1.

**Table 1 molecules-30-04328-t001:** Structural specifics, distribution, and key molecular targets and biological activities of representative ellagitannins.

Compound	Structural Specifics	Plant Species/Distribution	Key Molecular Mechanisms and Bioactivities	Refs.
**Sanguiin H-6**	Dimeric ellagitannin. casuarictin-type units, linked via a bond between a gallic acid residue and one of the HHDP acid moieties.	Found in many *Rubus* species (e.g., *Rubus coreanus*, *Rubus idaeus*, *Rubus fruticosus*). Also present in *Sanguisorba officinalis*, *Fragaria* (strawberries), *Alchemilla*, and others. Very widespread in Rosaceae.	Anticancer: induces apoptosis in breast cancer cells via caspase-8, caspase-3 activation, and modulation of Bax/Bcl-2 ratio. Activation of MAPK p38; truncated BID; in ovarian carcinoma. Irreversible inhibition of topo I/II catalytic function (prevents formation of covalent enzyme-DNA intermediates) inhibits bone resorption by suppressing RANKL-induced osteoclast differentiation, downregulation of NFATc1, c-Src, cathepsin K, c-Fos, NF-κB; inhibition Antibacterial activity: Inhibits biofilm formation by MRSA and reduces the growth of MRSA in vivo. General antioxidant, anti-inflammatory potentials via ROS scavenging, mitigation of ROS and GSH depletion, and modulation of NF-κB, as typical for ellagitannins. Anti-angiogenic via blocking VEGF binding to its receptor (VEGFR2/KDR).	[52,53,65,182,185,186,227,232]
**Corilagin**	Monomeric ellagitannin. Glucose core esterified with one galloyl + one HHDP (hexahydroxydiphenoyl) group. Also referred to as gallotannin.	Present in many plant families: Combretaceae (e.g., *Terminalia catappa*), Euphorbiaceae (e.g., *Phyllanthus* spp.), Geraniaceae, Polygonaceae, Saururaceae, etc. Found in aerial parts (leaves, fruits, seeds) of these plants.	Anti-cancer: modulation of STAT3/5, MAPK signaling in gastric cancer cells. Induces apoptosis via extrinsic (caspase-8) and intrinsic/mitochondrial pathway (caspase-9, Bax/Bcl-2 modulation). Induces autophagy (suppressing Akt/mTOR/p70S6K) in breast cancer (MCF-7). Cell cycle arrest at G2/M: downregulation of Cyclin B1, Myt1, Wee1, phospho-Cdc2, etc., in ovarian cancer model. Induces DNA damage; downregulates RNF8 (E3 ubiquitin ligase involved in DNA damage response), thereby impairing DNA repair. Anti-hyperalgesic (pain) effects in mice: in models like acetic acid writhing, formalin, capsaicin, glutamate, hot-plate; effectively reduces nociception relative to standard NSAIDs. Anti-inflammatory: inhibits LPS-induced inflammation (e.g., reduces NO production, reduces pro-inflammatory cytokines) in macrophage RAW264.7 cells.	[15,233,234,235,236,237,238,239]
**Geraniin**	Dehydroellagitannin (a more oxidized form) monomeric ellagitannin with additional modifications. It has a glucose core with HHDP, potentially dehydro-HHDP; capable of isomerization between hemi-ketal forms.	Found in many species: *Geranium* spp., *Phyllanthus* spp., rambutan rind (*Nephelium lappaceum*), etc. Distribution often in plants used in traditional medicine in Asia.	Antioxidant, free radical scavenging Anticancer: suppression of NF-κB activation, modulation of PI3K/Akt/mTOR pathways in HT-29 colorectal cancer model Enzyme inhibition: α-glucosidase Immunomodulatory effects: inducing cytokines, modulating macrophages yields gallic acid, hexahydroxydiphenic acid, and corilagin upon hydrolysis	[133,240,241,242]
**Punicalagin**	Large hydrolysable ellagitannin. Exists as α and β isomers. Glucose core esterified with galloyl and multiple HHDP moieties. Highest molecular weight among monomeric ellagitannins	Best known from *Punica granatum* (pomegranate): peel, juice, leaves, bark. Also present in *Terminalia catappa*, *Terminalia myriocarpa*, *Combretum molle* etc.	Anticancer: induces apoptosis, reduces invasive capacity (via MMPs, Snail, Slug) in gastric and colon cancer models Induction of apoptosis (via caspase-3, caspase-8, caspase-9 activation), upregulation of pro-apoptotic (Bax), downregulation of anti-apoptotic (Bcl-2) proteins. Autophagy induction: mTOR down-regulation, ULK1 up-regulation. Inhibition of NF-κB signaling: degradation of IκBα, reduced nuclear translocation of p65; decreased production of downstream IL-6, IL-8 inhibition of topo I and II, interfering with DNA relaxation. Anti-inflammatory and antioxidant properties: suppresses inflammatory cytokines, oxidative stress, activation of Nrf2/Keap1 pathway; downregulation of NF-κB. Hepatoprotective activity: protects against chemical-induced liver injury, enhances autophagy, etc. Enzyme inhibition: α-glucosidase inhibition, etc. Purified punicalagin from peel showed mixed inhibition of α-glucosidase.	[158,243,244,245,246,247,248,249,250,251,252]
**Oenothein B**	Macrocyclic dimeric ellagitannin: two monomeric ellagitannin units coupled in a macrocyclic dimer structure (ring closure by intermolecular coupling). Rigid structure with restricted flexibility	Isolated from *Oenothera erythrosepala*, *Oenothera biennis*, various *Epilobium* species. Widely distributed in Onagraceae, Myrtaceae, Lythraceae.	Immunomodulation: modulates cytokine production (IL-1β, TNF-α, IL-6, IL-8, IFN-γ, GM-CSF) by macrophages/monocytes/NK/γδ T cells; activation of phagocyte functions; NF-κB modulation. Dendritic cell modulation: suppresses DC differentiation/maturation; downregulates CD1a, CD83; Boosts host-mediated antitumor activity and prolongs survival in rodent models (activation of macrophages rather than direct cytotoxicity) Anti-inflammatory/antioxidant: inhibits ROS production, hyaluronidase, lipoxygenase, MPO, and neutrophil activation. Neuroprotective effects: activation of ERK2, CREB signaling in brain; suppression of neuroinflammation in mice. beneficial effects in prostatic hyperplasia via 5α-reductase inhibition	[58,60,75,91,92,93,94,97,219,253,254,255,256,257,258]
**Chebulagic acid**	Hydrolyzable benzopyran ellagitannin often co-occurs with chebulinic acid.	Major constituents in *Terminalia chebula* fruits; also present in *Terminalia bellirica*, *Terminalia arjuna*, and related *Terminalia* spp.	Ferroptosis inhibition: in rat bone marrow MSCs treated with erastin, chebulagic acid shows stronger inhibition than chebulinic acid; mechanism via ROS scavenging and iron chelation; HHDP moiety plays a key role. Anti-inflammatory activity: in vitro (RAW264.7 macrophages) and in vivo (LPS-induced with animal models) effects; suppressing inflammatory cytokines, COX inhibition Antiproliferative/anti-fibrotic properties: in hepatic stellate cells, suppresses TGF-β1-induced activation, reduces expression of Smad2,3,4, collagen I/III, PAI-1, etc. Both chebulagic and chebulinic acids act as influenza viral neuraminidase inhibitors	[37,59,259,260,261,262,263,264]
**Chebulinic acid**	A hydrolysable ellagitannin, structurally similar to chebulagic acid Has specific stereochemical features; exhibits different conformation (skew-boat) vs. chebulagic (chair) in computational studies.	Co-occurs with chebulagic acid in *Terminalia chebula* and related *Terminalia* species.	Ferroptosis inhibition: weaker than chebulagic acid in the same models; also via antioxidant/iron chelation mechanisms; conformation and HHDP involvement matter. Anti-inflammatory: similar to chebulagic, though possibly lower potency depending on assay. Anti-fibrotic/anti-hepatic stellate proliferation: inhibits Smad signaling, collagen expression, etc.	[37,260,261,264]

## 5. Limitations and Future Perspectives

### 5.1. Pharmacological and Translational Challenges


**Stability and pharmacokinetic issues**


Solubility and permeability remain the major pharmacokinetic liabilities of ellagitannins. High molecular weight, extensive hydrogen-bonding capacity, and polar surface area severely limit passive diffusion across enterocyte membranes. Attempts to increase exposure by high oral dosing often founder on poor absorption and extensive first-pass metabolism or on unanticipated local gastrointestinal effects. Experimental permeability studies (e.g., on geraniin and its hydrolysis products) demonstrate low apparent permeability for intact ellagitannins, whereas smaller hydrolysis products (gallic and ellagic acids) and urolithins show improved transcellular passage. These physicochemical constraints underlie the contemporary shift from using crude extracts towards targeted delivery strategies (prodrugs, nanoencapsulation, colon-targeted systems) or towards direct administration of microbial metabolites when a systemic effect is desired [265,266].

After ingestion, ellagitannins rarely reach effective systemic concentrations; their absorption rates are low and highly variable, and their plasma half-lives are truncated by first-pass hepatic and microbial metabolism. The fate of these compounds is largely determined by extensive hydrolysis to ellagic acid and subsequent microbial transformation to urolithins—the actual bioactive metabolites in circulation [4,42,43,220]. For example, serum concentrations of ellagic acid and its precursor punicalagin observed in rodents were markedly lower than doses used in vitro. Interindividual variation in microbiota composition and metabolic capacity further exacerbates variability in urolithin production and pharmacodynamics, complicating pharmacokinetic prediction and clinical dosing [267,268]. In addition, the pharmacokinetic profile of ellagitannins appears to be complicated by the activity of efflux transporters, widely distributed in various tissues and organs (P-gp, MRPs, OATP, SGLT1). In a Caco-2 model, corilagin and its hydrolysis products, gallic acid and ellagic acid, all showed very low apparent permeability, consistent with poor intrinsic absorption. Uptake studies revealed that corilagin and gallic acid, in particular, were subject to pronounced efflux, which was markedly attenuated by inhibitors of P-glycoprotein and other multidrug-resistance proteins, indicating that these tannin-derived phenolics are substrates for intestinal efflux pumps. Ellagic acid, by contrast, appeared to rely partly on carrier-mediated influx via OATP and SGLT1, as its transport decreased in the presence of their inhibitors. These findings highlight efflux transporter phenomena as an additional variable in ellagitannins’ bioavailability, superimposed on their already low solubility and extensive metabolism [56].


**Extraction, characterization, and formulation issues**


As with other high-molecular-weight polyphenols, ellagitannins exhibit substantial chemical lability, particularly under neutral to basic pH and elevated temperature, owing to multiple ester linkages and HHDP (hexahydroxydiphenoyl) groups that are prone to hydrolysis and oxidation. This instability significantly complicates their extraction, purification, and structural characterization. For example, sanguiin H-6 and lambertianin C degrade rapidly at pH 8, with half-lives dropping to ~2.5 h at 20 °C and even shorter at elevated temperatures [269]. Under more alkaline conditions (pH 10–11), many hydrolysable tannins display half-lives of under 10 min, driven by structural alterations where oxidation of phenolic groups can generate reactive quinones [270]. Similarly, studies have shown that under alkaline conditions, gallic acid undergoes oxidative dimerization, forming purpurogallin-8-carboxylic acid, which contributes to marked inhibition of xanthine oxidase activity. This illustrates how pH-dependent structural transformations can both compromise stability and, in some cases, yield new bioactive metabolites with distinct pharmacological properties [271].

In addition, isolation yields of ellagitannins can be low, and structural diversity (oligomerization, inter-ester and ether bonds, variable galloyl, HHDP and DHHDP substitutions) further complicates purification and can impede scale-up for clinical development. The formidable macrocyclic and oligomeric structures of some ellagitannins (such as oenothein B and sanguiin H-6) can pose analytical challenges, leading to signal overlap and line broadening in NMR spectroscopy and difficulties in purification and characterization [60].


**Toxicity and safety considerations**


Ellagitannins are generally well-tolerated in preclinical models. For instance, chebulagic acid administered at doses up to 2000 mg/kg in Sprague-Dawley rats produced no observable toxicity [272]. Oenothein B, while not genotoxic, exhibited cytotoxic effects only following prolonged high-dose exposure over 48 h [273]. Some compounds, such as geraniin, may cause mild astringency that can reduce oral acceptability; nevertheless, overall toxicity in animal studies remains low, supporting their relative safety in preclinical contexts [134].


**Limited clinical evidence**


Although large-scale clinical trials remain limited, clinical evidence for ellagitannins, ellagic acid, and their gut microbiota-derived metabolites (urolithins) in humans has been emerging across trials focused on metabolic syndrome, inflammation, muscle health, and cancer risk.

One representative randomized clinical trial tested whole pomegranate juice (rich in ellagitannins) in patients with inflammatory bowel disease (IBD). Over 12 weeks, regular intake of pomegranate juice led to significant changes in the gut microbiome and reduced fecal calprotectin, a surrogate marker for mucosal improvement, along with altered plasma cytokine profiles. The study also monitored urinary and plasma ellagitannin metabolites, confirming systemic uptake and bioconversion in humans [274].

In prostate cancer, a 4-week clinical trial examined black raspberry nectar and confectionary products standardized for ellagitannin content in men prior to prostatectomy. While focused on feasibility and safety, the trial detected dose-dependent increases in urinary urolithin A and dimethyl ellagic acid as biomarkers of ellagitannin metabolism, supporting the functional bioconversion of these polyphenols even in ill patients. These formulations have since been proposed for future cancer prevention trials [275].

A systematic review and meta-analysis of randomized controlled trials assessed the impact of ellagitannin-rich fruit consumption on blood pressure and vascular outcomes. Five randomized controlled trials collectively found that fruit intervention lowered diastolic blood pressure and improved endothelial function, although effects on systolic blood pressure were less robust. This reflects an anti-inflammatory and vasoprotective clinical fingerprint, consistent with mechanistic expectations [276].

Significant recent progress has come from trials on urolithin A, the direct microbial metabolite of ellagic acid. In a designed placebo-controlled study, older adults (mean age 71.7 years) given oral urolithin A daily for up to 4 months demonstrated measurable improvement in muscle endurance and mitochondrial function markers, with lower blood ceramide and C-reactive protein compared to placebo. Notably, adverse events did not differ between groups, confirming safety [277].

Additional clinical trials have shown positive effects for ellagic acid in metabolic syndrome and diabetes. Daily administration of 180 mg ellagic acid for 8 weeks led to modest but statistically significant reductions in some lipid profile variables and improved antioxidant status in adults, without serious adverse events [278]. Similar benefits have been reported for glycemic control, insulin resistance, and inflammatory markers, especially among patients with chronic disease backgrounds [279].

### 5.2. Emerging Research Perspectives


**Alternative routes and urolithins administration**


Given their physicochemical and metabolic limitations, parenteral delivery strategies for ellagitannins and their bioactive metabolites are being actively investigated. Current preclinical studies have primarily employed oral and intraperitoneal routes for the administration of urolithins, bypassing microbial conversion, with urolithin A receiving particular attention. Evidence indicates that it can adequately cross the BBB, supporting its potential for targeting neurological conditions. Moreover, urolithin A displays a favorable pharmacokinetic profile: it is bioavailable across tested doses, does not accumulate over time, and demonstrates a relatively long half-life (17–22 h), most likely due to active enterohepatic recirculation. Administration routes should be optimized according to disease specifics, localization of organ and tissue targets, and pathophysiological context [280].

From a translational perspective, human studies on urolithin A remain limited and have thus far centered on its effects on mitochondrial and muscle health. A Phase I clinical trial confirmed the activation of mitochondrial biomarkers in both muscle and plasma, consistent with earlier findings in cell-based and in vivo models [280,281].


**Chemical modifications and formulation approaches**


Chemical modification, prodrug and conjugation strategies, as well as formulation approaches, can offer substantial solutions to the bioavailability and delivery challenges faced by ellagitannins and their hydrolysis products.

For example, heat processing of strawberry purée minimally impacts the bioconversion of ellagitannins to urolithins by gut microbiota, as similar urolithin excretion profiles have been observed between fresh strawberries and thermally treated purées with equivalent ellagitannin content [282]. While processing may increase free ellagic acid, the type of food matrix and accessibility of ellagitannins do not substantially alter the microbial transformation to urolithins. Polyphenols generally occur in foods as esters, glycosides, and polymers, forms that require extensive hydrolysis by gastrointestinal enzymes and/or microbiota for absorption. It is estimated that about 48% of dietary polyphenols are transformed in the small intestine, 42% in the large intestine, and approximately 10% persist undigested [283].

Bioaccessibility of ellagic acid significantly increases throughout the digestive phases, rising during intestinal digestion due to progressive release from ellagitannins and food matrices, a process potentially facilitated by bile salts and pancreatin. Studies in diverse fruits confirm that liberation of ellagic acid is promoted by gastrointestinal digestion, enhancing its availability for microbial conversion and absorption [283]. Advances in delivery technology, notably the use of supersaturatable self-microemulsifying drug delivery systems (S-SMEDDS), have markedly increased the oral bioavailability of ellagic acid. Animal pharmacokinetic studies report a 4.7- to 5.8-fold higher area under the curve (AUC) with S-SMEDDS compared to suspensions, and reduced clearance rates [284].

Numerous micro- and nanotechnology-based delivery systems, including microspheres, nanoparticles, pH-sensitive microassemblies, and metalla-cages, have been developed to address the poor solubility and bioavailability of ellagic acid. These systems enhance aqueous solubility, membrane permeability, and targeted delivery, particularly for anticancer applications. Encapsulation methods, such as polymer-based nanoparticles, cyclodextrin complexes, and thermosensitive liposomes, have been shown to significantly elevate systemic ellagic acid concentrations, facilitate tumor-specific delivery, and potentiate cytotoxic effects on cancer cells [285].

Despite these advances, the oral bioavailability of ellagic acid remains limited by its poor solubility and membrane permeability, a consequence of its rigid structure and abundant hydroxyl groups. These characteristics classify ellagic acid as a class IV compound in the biopharmaceutical classification system, restricting its ability to reach therapeutic levels in plasma and tissues. Next-generation delivery technologies, including nanoformulations and surface-modified nanoparticles, offer promising strategies for overcoming these barriers by improving intestinal uptake and enabling lymphatic transport, thereby optimizing systemic exposure and therapeutic efficacy [286].

Multiple studies have now established that encapsulation in polymeric nanoparticles can significantly improve the oral bioavailability and therapeutic outcomes of ellagitannins and their derivative, ellagic acid. For example, poly(ε-caprolactone) nanoparticles produced by emulsion-diffusion-evaporation have been used to load ellagic acid, resulting in a 3.6-fold increase in area under the plasma concentration-time curve in rabbit models compared to free forms; oral administration protocols are typically utilized, with interventions lasting several weeks to permit both pharmacokinetic and efficacy assessments. Similarly, micro- and nanodevices using biocompatible polymers such as chitosan, soy phosphatidylcholine, maltodextrin, β-cyclodextrin, and PEGylated liposomes have been employed to provide improved stability, uptake, and retention in rodent and rabbit in vivo models, reflecting a move toward both systemic and tissue-targeted formulations. Chitosan nanoparticles, for instance, leverage their positive surface charge and biocompatibility to enhance site-specific delivery of ellagic acid to target organs, with nanoparticle sizes in the 20–60 nm range proving effective in cell models and animal studies. Routes of administration vary as appropriate to the pharmacological target, but oral, intravenous, and intraperitoneal interventions predominately prevail, depending on the tissue distribution and bioavailability objectives sought [287,288,289,290].

Intervention periods for these stabilization strategies typically range from single-dose pharmacokinetic analyses up to multi-week or even month-long regimens when evaluating sustained efficacy, antitumor, or organ-protective effects. Notably, advanced delivery devices such as pH-sensitive exosomes and dendrimer nanoparticles have now demonstrated enhanced brain delivery of urolithins in neuronal tumor models, while PEGylated liposomal platforms have been shown to prolong systemic circulation and facilitate tissue targeting for chronic disease applications [286].

Furthermore, recent advancements in micro- and nanodevices have garnered attention as promising drug delivery systems for ellagic acid, designed to enhance its bioavailability, utilizing biocompatible materials such as biopolymers [285]. Formation of geraniin-soy phosphatidylcholine complexes has shown to significantly enhance oral bioavailability and plasma retention of active metabolites, as confirmed by higher entrapment and complexation efficiencies, superior physicochemical stability, and an 11-fold increase in ellagic acid exposure compared to free geraniin [290]. In addition, encapsulation of urolithins A, B, and IsoUro-A in milk-derived exosomes improves brain delivery efficiency by 3–4 times and achieves a marked increase in antiproliferative activity and cell death in neuronal tumor models versus free forms [291]. PEGylated liposome delivery has likewise boosted the circulation half-life, cellular uptake, and antitumor potency of urolithin A in vitro and in vivo [292].

Structural modifications, such as methylated urolithin A, produce a superior effect in vivo, mitigating neuroinflammation and oxidative damage and restoring mitochondrial function in aging models [293]. In addition, chemically conjugated urolithin A-NSAID prodrugs efficiently inhibit glucuronidation and improve mucosal bioavailability and tight junction integrity in intestinal cells, widening the therapeutic window of urolithin-based interventions [294].

Microbiome-targeted strategies are also gaining traction, such as pre- or probiotic co-administration to boost specific urolithin-producing taxa. Personalized microbiome profiling can identify metabotypes most likely to respond to ellagitannin supplementation, with potential for companion biomarker design in clinical studies [295].

## 6. Conclusions

In summary, this review consolidates the latest advances in understanding the structure, natural occurrence, and biological roles of representative ellagitannins—punicalagin, sanguiin H-6, corilagin, geraniin, oenothein B, chebulagic, and chebulinic acids, as well as their unique transformation into bioactive urolithins via gut microbial metabolism. Robust mechanistic and preclinical evidence highlights their antioxidant, anti-inflammatory, antimicrobial, and anticancer efficacy, spanning a broad spectrum of disease states. Nevertheless, core barriers such as poor bioavailability, variability in host microbiome metabolism, and limited clinical data still impede their full therapeutic translation. Encouragingly, recent advances are addressing these translational hurdles with novel approaches: microbiome-targeted interventions, innovative drug delivery systems, and metabolite-based strategies, all aiming to improve ellagitannin bioavailability and systemic exposure to urolithins and facilitate more consistent clinical outcomes.

## Figures and Tables

**Figure 1 molecules-30-04328-f001:**
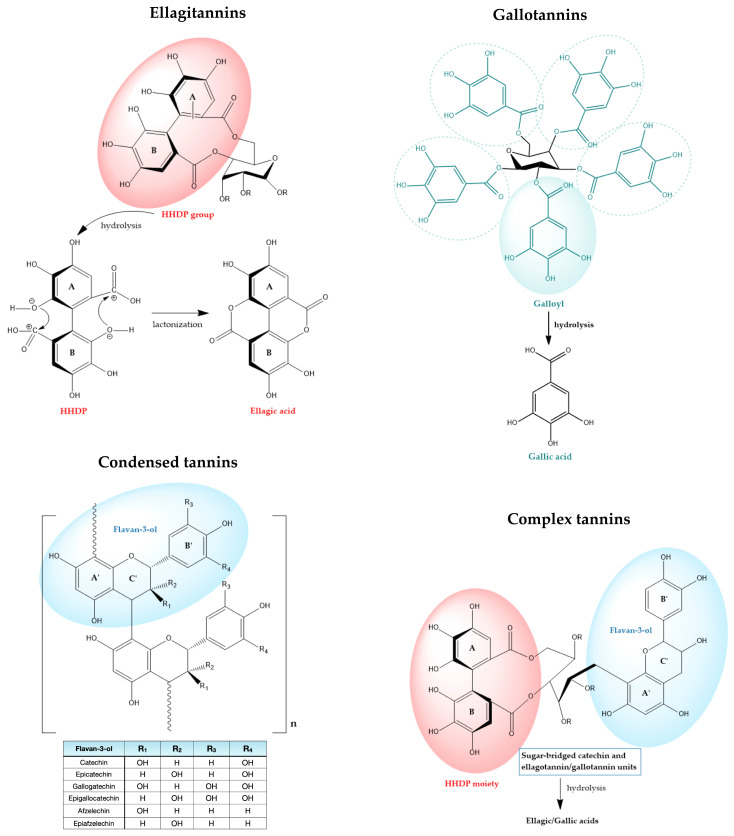
Overview of tannin classes and their representative structures. Key functional groups are circled. Aromatic rings of the HHDP group in ellagitannins are identified as A and B, with the flavan-3-ol rings correspondingly labeled A’, B’ and C’.

**Figure 2 molecules-30-04328-f002:**
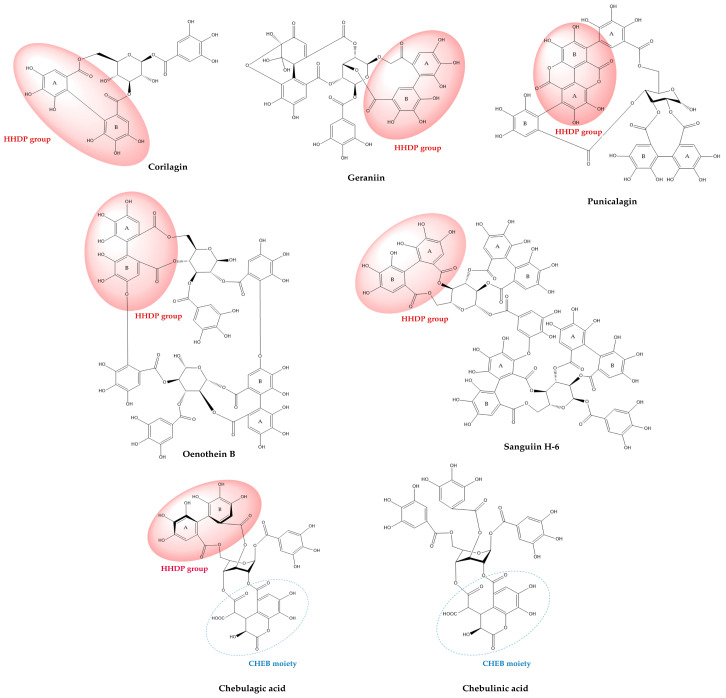
Chemical structures of selected ellagitannins, showing characteristic HHDP and chebuloyl groups.

## Data Availability

All data is available in the manuscript.

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
