# Peer review of "Unlocking the Therapeutic Potential of Ellagitannins: A Comprehensive Review of Key Representatives"

_molecules, 2025, doi:10.3390/molecules30224328_

Round 1

Reviewer 1 Report

Comments and Suggestions for Authors

The paper is interesting and presents a thorough review of the state of the art regarding the main ellagitannins found in nature. It offers a detailed description of the chemical characteristics of these compounds. However, to improve the clarity and depth of the manuscript, the following revisions are recommended:

  1. Include in vitro studies on the bioaccessibility of hydrolyzable tannins. There is recent literature describing their digestive behavior, which should be incorporated. The discussion should expand on the digestive physiology of these compounds and explain how this relates to their low bioavailability.
  2. Incorporate information about the influence of different food or pharmaceutical matrices on the fate of these compounds throughout the digestive system, and how their pharmacokinetics vary depending on the delivery vehicle. This topic is only superficially addressed in the current version.
  3. Improve the description of the studies and methodologies used to stabilize ellagitannins and enhance their bioavailability. The discussion on encapsulation or complex formation technologies is brief and lacks essential details such as the polymers used, doses, routes of administration, models applied, and intervention duration. This information is needed to provide a clearer understanding of the impact of these stabilization strategies.
  4. Strengthen the state of the art and discussion regarding the effect of ellagitannins on gut microbiota. The authors should describe how bacterial populations and diversity are modified, and relate these changes to health outcomes, especially the preventive effects on different types of gastrointestinal cancers. It is important to specify in which models these effects have been tested, the type and dose of ellagitannins used, the intervention period, and whether microbiota sequencing or diversity analyses were performed to substantiate the claims.
  5. Although the authors mention that human studies are scarce, it is important to include the existing ones in order to give more weight to the potential of these compounds for population health. A dedicated paragraph summarizing the available clinical evidence on supplementation with ellagitannin-rich foods or extracts is recommended.
  6. Add a description of the review’s literature search criteria, including years covered, databases consulted, and keywords used.
  7. Check formatting and nomenclature consistency. All scientific names should be italicized, and all compound names verified for accuracy (e.g., Line 47: “ellagotannins” should be corrected to “ellagitannins”).
  8. Clarify the differences in the amount of available evidence among ellagitannins. The manuscript currently gives the impression that data are comparable for all compounds, but most information pertains to urolithins and punicalagin, particularly in pharmaceutical, food, and biological contexts. The discussion should explicitly reflect this imbalance and emphasize where the strongest evidence lies.

Author Response

Responses to Reviewer 1

The paper is interesting and presents a thorough review of the state of the art regarding the main ellagitannins found in nature. It offers a detailed description of the chemical characteristics of these compounds. However, to improve the clarity and depth of the manuscript, the following revisions are recommended:

Comment 1: Include in vitro studies on the bioaccessibility of hydrolyzable tannins. There is recent literature describing their digestive behavior, which should be incorporated. The discussion should expand on the digestive physiology of these compounds and explain how this relates to their low bioavailability.
Response: Thank you for your valuable comment regarding the inclusion of in vitro studies on the bioaccessibility and digestive behavior of hydrolyzable tannins, and for emphasizing the need to expand the discussion on digestive physiology as it relates to their low bioavailability.

These aspects are comprehensively addressed in several sections of the revised manuscript:
The discussion on the physicochemical and metabolic limitations of hydrolyzable tannins is detailed in the section reviewing their structural diversity and structure-activity relationships (Section 3). Here, the manuscript explains that high molecular weight, extensive hydroxylation, polymerization, and the formation of macrocyclic and oligomeric structures result in poor membrane permeability and limited absorption in the native form. In Section 5.1. Pharmacological and translational challenges are further explored with evidence from in vitro permeability studies using Caco-2 cell models, showing that corilagin and its ts hydrolysis products gallic acid and ellagic acid all exhibit very low apparent permeability. The role of efflux transporters (e.g., P-gp, MRPs, OATP, SGLT1) in restricting absorption and promoting intestinal efflux is specifically discussed (lines 981-990).
These aspects have been further elaborated in the expanded Section 5.2 (Emerging research perspectives, Chemical modifications, and formulation approaches), which now discusses recent in vitro, in vivo, and clinical studies addressing the bioaccessibility and digestive behavior of hydrolysable tannins and their metabolites (lines 1076-1134).

Comment 2: Incorporate information about the influence of different food or pharmaceutical matrices on the fate of these compounds throughout the digestive system, and how their pharmacokinetics vary depending on the delivery vehicle. This topic is only superficially addressed in the current version.

Response: Thank you for your suggestion. In response to your recommendation, Section 5.2, “Emerging Research Perspectives: Chemical Modifications and Formulation Approaches”, has been substantially expanded to address how food and pharmaceutical matrices modulate the fate and pharmacokinetics of ellagitannin-derived compounds throughout the digestive system (lines 1076-1134).

Comment 3: Improve the description of the studies and methodologies used to stabilize ellagitannins and enhance their bioavailability. The discussion on encapsulation or complex formation technologies is brief and lacks essential details such as the polymers used, doses, routes of administration, models applied, and intervention duration. This information is needed to provide a clearer understanding of the impact of these stabilization strategies.

Response: Thank you for your recommendation. To address your comment, the manuscript has been expanded in Section 5.2 (“Emerging research perspectives - Chemical modifications and formulation approaches”). Here, recent advances in encapsulation technologies are clarified, emphasizing both the materials used and the relevant experimental parameters.

Comment 4: Strengthen the state of the art and discussion regarding the effect of ellagitannins on gut microbiota. The authors should describe how bacterial populations and diversity are modified, and relate these changes to health outcomes, especially the preventive effects on different types of gastrointestinal cancers. It is important to specify in which models these effects have been tested, the type and dose of ellagitannins used, the intervention period, and whether microbiota sequencing or diversity analyses were performed to substantiate the claims.

Response: Thank you for your insightful suggestion. In response, we have expanded our discussion in Section 3 (“Structural diversity and structure-activity relationships of representative ellagitannins”) to describe in detail how ellagitannin and ellagic acid intake modulate gut microbial populations, and how these alterations contribute to gastrointestinal health and cancer prevention (lines 199-228).

Comment 5: Although the authors mention that human studies are scarce, it is important to include the existing ones in order to give more weight to the potential of these compounds for population health. A dedicated paragraph summarizing the available clinical evidence on supplementation with ellagitannin-rich foods or extracts is recommended.

Response: Thank you for your note. Accordingly, the manuscript now includes a dedicated paragraph summarizing key findings from existing human studies on ellagitannin-rich foods and their metabolic products, providing real-world evidence of their health benefits and their growing relevance for inclusion in nutrition science and public health strategies (lines 1021-1055).

Comment 6: Add a description of the review’s literature search criteria, including years covered, databases consulted, and keywords used.

Response: Thank you for the comment. We are pleased to provide a clear description of the literature search criteria for the revised manuscript, noting that this is a narrative review rather than a systematic review. The search covered publications from the year 2000 to August 2025 and primarily utilized databases including PubMed/MEDLINE, Scopus, and Web of Science. Complementary searches were conducted via Google Scholar to capture recent articles and in-press manuscripts. The search keywords included the names of the key representative ellagitannins (punicalagin, sanguiin H-6, geraniin, corilagin, oenothein B, chebulagic acid, and chebulinic acid), as well as more general terms such as “ellagitannins,” “urolithins,” “anti-inflammation,” “immunomodulation,” “anticancer,” “metabolic diseases,” “antioxidant,” and descriptors like “in vitro”, “in vivo” and “clinical” studies. Boolean operators were applied to refine and broaden the search, with studies selected based on relevance and quality. References cited in key articles were also reviewed to ensure comprehensive coverage.

Comment 7: Check formatting and nomenclature consistency. All scientific names should be italicized, and all compound names verified for accuracy (e.g., Line 47: “ellagotannins” should be corrected to “ellagitannins”).

Response: We appreciate your suggestion. The formatting and nomenclature throughout the manuscript have been thoroughly reviewed and corrected for consistency. All scientific names are now italicized according to standard conventions, and a typographical error in “ellagotannins” (Line 46 in the revised manuscript) has been corrected to “ellagitannins.”

Comment 8: Clarify the differences in the amount of available evidence among ellagitannins. The manuscript currently gives the impression that data are comparable for all compounds, but most information pertains to urolithins and punicalagin, particularly in pharmaceutical, food, and biological contexts. The discussion should explicitly reflect this imbalance and emphasize where the strongest evidence lies.

Response: We thank the reviewer for this insightful comment. We have revised the manuscript to clarify the imbalance in the amount of available evidence among ellagitannins and their metabolites. The revised text now emphasizes that most data are concentrated on urolithins and punicalagin, particularly regarding their pharmacological, nutritional, and biological effects, while other ellagitannins remain comparatively less studied (lines 88-95, 295-306).

Reviewer 2 Report

Comments and Suggestions for Authors

The reviewed article is an attempt to discuss the occurrence and health-promoting properties of selected ellagitannins.

Ellagitannins are a very important group of polyphenols, and in recent years, increasing attention has been paid to studying their structure and properties. However, ellagitannins are a challenging subject for research. Their complex structure, structural diversity, chemical reactivity, and ability to form complexes with other components make the investigation of their properties a difficult task.

 The authors focused their attention on several ellagitannins: punicalagin, sanguiin H-6, corilagin, geraniin, oenothein B, chebulagic, and chebulinic acids. There is a lack of explanation as to why the authors selected these particular ellagitannins as the subject of their literature review. It might be worth specifying the criteria for selecting the above-mentioned ellagitannins more clearly.

The authors focused on collecting literature data concerning the antioxidant and anti-inflammatory activities, antimicrobial properties, therapeutic potential in cardiovascular and metabolic disorders, anticancer activity, neuroprotective properties, and anti-osteoporotic effects of ellagitannins.The article lacks data on the structure of the discussed ellagitannins. There is not even a mention of their molar mass. A valuable addition would be a discussion of the structure of selected ellagitannins, including information on their fragmentation and related characteristics.

In addition, the article also addresses stability and pharmacokinetic issues as well as aspects related to the extraction, characterization, and formulation of ellagitanninsAlthough these topics are discussed only briefly.

Nevertheless, the article is well written and constitutes a valuable review of the literature concerning the above-mentioned ellagitannins.

Below my detailed comments:

Line 15: In the literature, tannins are classified into hydrolysable and condensed tannins. Hydrolysable tannins include ellagitannins and gallotannins. I have never encountered the term “hydrolysable ellagitannins” used by the authors. I believe that the terminology currently in use should be followed, and the term “hydrolysable tannins” should be applied.

Lines 23, 70, 178, 186, 236: “In vivo” should be written in italics.

Line 29: punicalagin instead of “pinicalagin”.

Line 41: Please indicate a scientific publication in which the term “hydrolysable polyphenolic tannins” appears. The reviewer has not encountered this term before. As I mentioned earlier, the term “hydrolysable tannins” is the one commonly used in the literature.

Lines 79, 113, : I believe that the term “hydrolysable tannins” or “ellagitannins” should be used instead of the term “hydrolysable ellagitannins” applied by the authors.

Line 81: Ellagitannins are present in all parts of plants, including stems, flowers, roots, rhizomes, and others.

Line 124: In the passage covering verses 46–48, complex tannins are classified as hydrolysable tannins, whereas in verse 124 they are classified simply as tannins. Please verify the classification of complex tannins in the literature.

Line 144: Please indicate a scientific publication on the basis of which the authors use the term “complex hydrolysable tannins.”

Lines 190-191: The molecular weight of punicalagin is 1084.72. Unfortunately, I did not find this information in the article, which is a pity. Therefore, it is not true that punicalagin “is among the highest molecular-weight ellagitannins identified in food”. For ellagitannins, this is not a particularly high molecular weight.

Line 198-199: Please specify in which solvent the solubility is meant.

Lines 204: “In vitro” should be written in italics.

Line 943: Names of ellagitannins, like those of any other chemical compounds, are written in lowercase.

Line 954: Please verify the correctness of the abbreviation HDPD.

The references (277 items) are well suited to the topics discussed. Most of them come from the last 20 years. Some older sources are also included; however, their use is justified, as they provide an important contribution to research on ellagitannins.

Author Response

Responses to Reviewer 2

The reviewed article is an attempt to discuss the occurrence and health-promoting properties of selected ellagitannins.

Ellagitannins are a very important group of polyphenols, and in recent years, increasing attention has been paid to studying their structure and properties. However, ellagitannins are a challenging subject for research. Their complex structure, structural diversity, chemical reactivity, and ability to form complexes with other components make the investigation of their properties a difficult task.

 Comment 1: The authors focused their attention on several ellagitannins: punicalagin, sanguiin H-6, corilagin, geraniin, oenothein B, chebulagic, and chebulinic acids. There is a lack of explanation as to why the authors selected these particular ellagitannins as the subject of their literature review. It might be worth specifying the criteria for selecting the above-mentioned ellagitannins more clearly.
Response: Thank you for your comment. In response, we have revised the aims section to clarify the criteria for selecting the seven representative ellagitannins reviewed (lines 65-67). Our selection was based on their structural diversity, prominence as major bioactive constituents in medicinal and dietary plants, and their frequent citation in the scientific literature as archetypal or functionally significant hydrolyzable ellagitannins. These compounds (punicalagin, sanguiin H-6, geraniin, corilagin, oenothein B, chebulagic acid, and chebulinic acid) collectively represent key scaffolds within the structural and biological landscape of the ellagitannin family, capturing both well-studied and under-explored analogues that reflect the current balance of evidence and highlight research gaps.

Comment 2: The authors focused on collecting literature data concerning the antioxidant and anti-inflammatory activities, antimicrobial properties, therapeutic potential in cardiovascular and metabolic disorders, anticancer activity, neuroprotective properties, and anti-osteoporotic effects of ellagitannins. The article lacks data on the structure of the discussed ellagitannins. There is not even a mention of their molar mass. A valuable addition would be a discussion of the structure of selected ellagitannins, including information on their fragmentation and related characteristics.
Response: Thank you for your comment. We have addressed the structural features of the representative ellagitannins in Section 3, “Structural diversity and structure-activity relationships (SARs) of representative ellagitannins,” (lines 148-158; 229-290) where we discuss their molecular architecture, characteristic functional groups, and variations in macrocyclic, monomeric, and dimeric forms. Hydrolysis patterns and unique chemical properties of each compound are outlined for punicalagin, sanguiin H-6, corilagin, geraniin, oenothein B, chebulagic acid, and chebulinic acid. Figure 1 presents the core tannin classes and signature moieties, while Figure 2 provides chemical structures of the selected ellagitannins, highlighting the diversity and complexity of their HHDP, galloyl, and chebuloyl groups. This section also clarifies that the primary active principles of dietary ellagitannins are their hydrolysis product, ellagic acid, and, predominantly, the gut microbiota-derived urolithins.

Comment 3: In addition, the article also addresses stability and pharmacokinetic issues as well as aspects related to the extraction, characterization, and formulation of ellagitanninsAlthough these topics are discussed only briefly.
Response: Thank you for your comment. Section 5.1 ("Pharmacological and translational challenges", Stability and pharmacokinetic issues) provides a synthesis of the current state of these topics, highlighting the major obstacles such as low solubility, poor membrane permeability, variable absorption, and extensive metabolism following ingestion, which have also been discussed in Section 3. “Structural diversity and structure-activity relationships (SARs) of representative ellagitannins”. It is emphasized throughout the review that available pharmacokinetic data on intact ellagitannins are limited because most in vivo biological effects are mediated by their hydrolysis product, ellagic acid, and, more importantly, by the gut-derived urolithins. Additionally, Section 5 has been expanded to more thoroughly address recent advancements in formulation strategies and prodrug design intended to enhance stability and systemic exposure, while the chemical lability challenges are detailed in the subsections dealing with structural characterization (Sections 3).

Comment 4: Nevertheless, the article is well written and constitutes a valuable review of the literature concerning the above-mentioned ellagitannins.
Response: Thank you very much for your positive feedback. We appreciate your recognition of the manuscript’s quality and value and are grateful for your thoughtful assessment of our review.

Below my detailed comments:

Comment 5: Line 15: In the literature, tannins are classified into hydrolysable and condensed tannins. Hydrolysable tannins include ellagitannins and gallotannins. I have never encountered the term “hydrolysable ellagitannins” used by the authors. I believe that the terminology currently in use should be followed, and the term “hydrolysable tannins” should be applied.
Response: Thank you for raising this point regarding terminology. With the use of the term “hydrolysable ellagitannins,” we intended to specifically distinguish ellagitannins from other hydrolysable tannins, namely gallotannins, as both groups share the capacity for hydrolysis but differ fundamentally in their core structures and biological properties. We use “hydrolysable ellagitannins” to clearly indicate we are referring to those hydrolysable tannins whose structures are defined by hexahydroxydiphenoyl (HHDP) groups and the capacity to yield ellagic acid upon hydrolysis, as opposed to gallotannins, which yield gallic acid and consist of linear or branched galloyl esters of glucose or related sugars.​
To address your concern and for transparency, we have included a list of references from primary research articles and scientific reviews where the term “hydrolysable ellagitannins” is used to refer to this specific subclass of tannins. Nevertheless, we have revised and simplified the phrasing to “ellagitannins”.
1. Valenti B, Natalello A, Vasta V, et al. Effect of different dietary tannin extracts on lamb growth performances and meat oxidative stability: comparison between mimosa, chestnut and tara. animal. 2019;13(2):435-443. doi:10.1017/S1751731118001556

  1. Mostafa Gouda, Laila Hussein, Douglas W. Wilson, Harpal S. Buttar. Multiple Therapeutic Applications of Pomegranate Fruit and its Bioactive Phytochemicals in Health and Disease. Biomedical Research, Medicine, and Disease, 1st Edition, 2023. DOI:10.1201/9781003220404-43.
  2. Haslam, E. (1992). Gallic Acid and Its Metabolites. In: Hemingway, R.W., Laks, P.E. (eds) Plant Polyphenols. Basic Life Sciences, vol 59. Springer, Boston, MA. https://doi.org/10.1007/978-1-4615-3476-1_10
  3. Christina Fjæraa Alfredsson, Menglei Ding, Qiu-Li Liang, Birgitta E. Sundström, Eewa Nånberg, Ellagic acid induces a dose- and time-dependent depolarization of mitochondria and activation of caspase-9 and -3 in human neuroblastoma cells, Biomedicine & Pharmacotherapy, Volume 68, Issue 1, 2014, Pages 129-135, ISSN 0753-3322. https://doi.org/10.1016/j.biopha.2013.08.010.
  4. Piteesha Ramlagan, Rola M. Labib, Mohamed A. Farag, Vidushi S. Neergheen,

Advances towards the analysis, metabolism and health benefits of punicalagin, one of the largest ellagitannin from plants, with future perspectives, Phytomedicine Plus, Volume 2, Issue 3, 2022, 100313, ISSN 2667-0313, https://doi.org/10.1016/j.phyplu.2022.100313.

  1. Alexova, R.; Alexandrova, S.; Dragomanova, S.; Kalfin, R.; Solak, A.; Mehan, S.; Petralia, M.C.; Fagone, P.; Mangano, K.; Nicoletti, F.; et al. Anti-COVID-19 Potential of Ellagic Acid and Polyphenols of Punica granatum L. Molecules 2023, 28, 3772. https://doi.org/10.3390/molecules28093772.

Comment 6: Lines 23, 70, 178, 186, 236: “In vivo” should be written in italics.
Response: Thank you for your observation. Although the journal style guidelines do not require these terms to be italicized, we have carefully reviewed the manuscript and have accordingly italicized all instances of “in vivo” and “in vitro” to ensure compliance with accepted scientific writing conventions.

Comment 7: Line 29: punicalagin instead of “pinicalagin”.
Response: Thank you for noticing the typographical error. “pinicalagin” has been replaced with “punicalagin”.

Comment 8: Line 41: Please indicate a scientific publication in which the term “hydrolysable polyphenolic tannins” appears. The reviewer has not encountered this term before. As I mentioned earlier, the term “hydrolysable tannins” is the one commonly used in the literature.
Response: Thank you for your question about terminology. Our use of the phrase “hydrolysable polyphenolic tannins” was intended to emphasize that these compounds are not only hydrolysable tannins, but also structurally and functionally belong to the broader class of dietary polyphenols. However, we recognize that “hydrolysable tannins” is the standard and widely accepted term in the scientific literature and have accordingly revised the sentence to avoid any confusion (line 40).

Comment 9: Lines 79, 113, : I believe that the term “hydrolysable tannins” or “ellagitannins” should be used instead of the term “hydrolysable ellagitannins” applied by the authors.
Response: Thank you for your note. The terminology has been revised accordingly in the manuscript. The term “hydrolysable ellagitannins” has been replaced with “ellagitannins” in the indicated sections.

Comment 10: Line 81: Ellagitannins are present in all parts of plants, including stems, flowers, roots, rhizomes, and others.
Response: Although we did not fully understand the intended suggestion, we have revised the sentence to clarify that ellagitannins are present in all parts of plants, providing a more accurate and comprehensive description (lines 81-82).

Comment 11: Line 124: In the passage covering verses 46–48, complex tannins are classified as hydrolysable tannins, whereas in verse 124 they are classified simply as tannins. Please verify the classification of complex tannins in the literature.
Response: We are happy to clarify this point. Upon reviewing, there is no misreference: in both lines 46–48 (lines 45-47 in the revised manuscript), as well as line 124 (line 153 in the revised manuscript), complex tannins are classified as hydrolysable tannins, specifically as “complex tannins (fusing hydrolysable and condensed motifs).” We have ensured consistency throughout the manuscript to avoid any confusion.

Comment 12: Line 144: Please indicate a scientific publication on the basis of which the authors use the term “complex hydrolysable tannins.”
Response: We appreciate the reviewer’s comment and have provided an appropriate reference supporting the use of the term “complex hydrolysable tannins.”
“Taking into account, for instance, the structural features of hydrolysable tannins, they can be identified as follows: gallotannins, ellagitannins and complex tannins.” - doi: 10.1111/bph.13630.
We used the term to specifically differentiate “complex tannins,” which are also hydrolysable by nature, from the two well-established subclasses - ellagitannins and gallotannins. We intended to clarify the classification and highlight structural distinctions within hydrolysable tannins.  If our phrasing caused confusion, we appreciate the suggestion and agree to refine the terminology in the updated manuscript to “complex tannins (hydrolysable flavono-ellagitannins)”, line 153.

Comment 12: Lines 190-191: The molecular weight of punicalagin is 1084.72. Unfortunately, I did not find this information in the article, which is a pity. Therefore, it is not true that punicalagin “is among the highest molecular-weight ellagitannins identified in food”. For ellagitannins, this is not a particularly high molecular weight.
Response: Thank you for your observation. To clarify, the manuscript refers to punicalagin as among the highest molecular-weight monomeric ellagitannins identified in food, with a molecular weight of 1084.72 g/mol. We agree that much larger oligomeric and polymeric ellagitannins exist, and we have revised the text accordingly to ensure accuracy. The molecular weight has now been reported explicitly, and the description has been specified to reflect punicalagin’s status as a high-molecular-weight monomer among ellagitannins found in dietary sources (lines 229-231).
Piteesha Ramlagan, Rola M. Labib, Mohamed A. Farag, Vidushi S. Neergheen. Advances towards the analysis, metabolism and health benefits of punicalagin, one of the largest ellagitannin from plants, with future perspectives, Phytomedicine Plus, Volume 2, Issue 3, 2022, 100313, ISSN 2667-0313. https://doi.org/10.1016/j.phyplu.2022.100313.

Comment 13: Line 198-199: Please specify in which solvent the solubility is meant.
Response: Thank you for the comment. We have clarified the solvent medium to which the solubility statement refers in the revised manuscript (lines 199–200).

Comment 14: Lines 204: “In vitro” should be written in italics.
Response: Thank you for your note. The term in vitro has been corrected and is now consistently written in italics throughout the manuscript.

Comment 15: Line 943: Names of ellagitannins, like those of any other chemical compounds, are written in lowercase.
Response: Thank you for your comment. We have reviewed the manuscript and corrected the names of all ellagitannins to be written in lowercase, in accordance with chemical nomenclature conventions, except when they appear at the beginning of a sentence.

Comment 16: Line 954: Please verify the correctness of the abbreviation HDPD.
Response: Thank you for noticing the typo. We have corrected the used abbreviation (HHDP and DHHDP).

Comment 17: The references (277 items) are well suited to the topics discussed. Most of them come from the last 20 years. Some older sources are also included; however, their use is justified, as they provide an important contribution to research on ellagitannins.
Response: Thank you very much for your kind assessment. We appreciate your recognition of the breadth and relevance of the reference list, as well as your understanding of the inclusion of foundational older sources that remain significant for research on ellagitannins.

Round 2

Reviewer 1 Report

Comments and Suggestions for Authors

The paper has satisfactorily addressed the reviewers’ comments, represents a valuable contribution to the discipline, and meets the requirements for publication